# Atmospheric organic vapors in two European pine forests measured by a Vocus PTR-TOF: insights into monoterpene and sesquiterpene oxidation processes

Haiyan Li[1], Manjula R. Canagaratna[2], Matthieu Riva[3], Pekka Rantala[1], Yanjun Zhang[1], Steven Thomas[1], Liine Heikkinen[1], Pierre-Marie Flaud[4,5], Eric Villenave[4,5], Emilie Perraudin[4,5], Douglas Worsnop[2], Markku Kulmala[1], Mikael Ehn[1], Federico Bianchi[1]

[1] Institute for Atmospheric and Earth System Research/Physics, Faculty of Science, University of Helsinki, Helsinki, 00014, Finland
[2] Aerodyne Research Inc., Billerica, Massachusetts 01821, USA
[3] Univ. Lyon, Université Claude Bernard Lyon 1, CNRS, IRCELYON, 69626, Villeurbanne, France
[4] University of Bordeaux, EPOC, UMR 5805 CNRS, 33405 Talence Cedex, France
[5] CNRS, EPOC, UMR 5805 CNRS, 33405 Talence Cedex, France

Correspondence: Haiyan Li (haiyan.li@helsinki.fi)

**Abstract.**

Atmospheric organic vapors play essential roles in the formation of secondary organic aerosol. Source identification of these vapors is thus fundamental to understand their emission sources and chemical evolution in the atmosphere and their further impact on air quality and climate change. In this study, a Vocus proton-transfer-reaction time-of-flight mass spectrometer (PTR-TOF) was deployed in two forested environments, the Landes forest in southern France and the boreal forest in southern Finland, to measure atmospheric organic vapors, including both volatile organic compounds (VOCs) and their oxidation products. For the first time, we performed binned positive matrix factorization (binPMF) analysis on the complex mass spectra acquired with the Vocus PTR-TOF and identified various emission sources as well as oxidation processes in the atmosphere. Based on separate analysis of low- and high-mass ranges, fifteen PMF factors in the Landes forest and nine PMF factors in the Finnish boreal forest were resolved, showing a high similarity between the two sites. Particularly, terpenes and various terpene reaction products were separated into individual PMF factors with varying oxidation degrees, such as lightly oxidized compounds from both monoterpene and sesquiterpene oxidation, monoterpene-derived organic nitrates, and monoterpene more oxidized compounds. Factors representing monoterpenes dominated the biogenic VOCs in both forests, with less contributions from the isoprene factors and sesquiterpene factors. Factors of the lightly oxidized products, more oxidized products, and organic nitrates of monoterpenes/sesquiterpenes accounted for 8-12% of the measured gas-phase organic vapors in the two forests. Based on the interpretation of the results relating to oxidation processes, further insights were gained regarding monoterpene and sesquiterpene reactions. For example, a strong relative humidity (RH)-dependence was found for the behavior of sesquiterpene lightly oxidized compounds. High concentrations of these compounds only occur at high RH, yet similar behavior was not observed for monoterpene oxidation products.

# 1 Introduction

Volatile organic compounds (VOCs) and their oxidation products are important contributors to atmospheric secondary organic aerosol (SOA) (Hallquist et al., 2009; Ehn et al., 2014) and new particle formation (Bianchi et al., 2016; Kirkby et al., 2016). Therefore, the identification of these organic vapors and their sources is fundamental for understanding the effects of atmospheric aerosols on climate change and air quality (Schell et al., 2001; Maria et al., 2004). Large amounts of VOCs with varying physicochemical properties are emitted from both biogenic and anthropogenic sources (Friedrich et al., 1999;

Kesselmeier et al., 1999), and their oxidation processes in the atmosphere can lead to the formation of thousands of structurally distinct products containing multiple functional groups (Hallquist et al., 2009; Wennberg et al., 2018). Due to the enormous challenge in characterizing these organic vapors at molecular level, knowledge of their sources or formation pathways has remained lacking.

Globally, SOA production from biogenic sources is much larger than that from anthropogenic sources (Tsigaridis and

45 Kanakidou, 2003). As a group of highly reactive gases, typically with one or more $C = C$ double bounds, terpenes make up a major fraction of biogenic VOCs, including isoprene, monoterpenes, and sesquiterpenes (Guenther et al., 1995). In the atmosphere, they react with various oxidants, i.e., hydroxyl radical (OH), ozone ($O_3$), and nitrate radical ($NO_3$), and produce a large variety of oxygenated molecules. Isoprene is the most emitted biogenic VOC on the global scale but has a relatively small SOA yield (Ahlberg et al., 2017; McFiggans et al., 2019). Monoterpenes are important sources of SOA (Ehn et al., 2014;

Zhang et al., 2018) and their oxidation processes have been found to play important roles in new particle formation (Kirkby et al., 2016; Simon et al., 2020). High ambient concentrations of monoterpenes have been observed in numerous pine forests (Hakola et al., 2012; Noe et al., 2012; Bourtsoukidis et al., 2014). While the concentrations of sesquiterpenes are generally much smaller than those of isoprene and monoterpenes (Sakulyanontvittaya et al., 2008; Sindelarova et al., 2014), sesquiterpenes could contribute significantly to SOA formation because of their reactivity and high aerosol yields (Calogirou

et al., 1999; Khan et al., 2017). Previous studies found that sesquiterpenes contributed to the $O_3$ and OH reactivity in forest environments (Kim et al., 2011; Hellén et al., 2018). The recently developed Vocus proton-transfer-reaction time-of-flight mass spectrometry (PTR-TOF) enables the real-time detection of both VOCs and their oxidation products. With a new chemical ionization source called Vocus, the instrument significantly improves its detection efficiency of product ions compared with conventional PTR instruments (Krechmer et al., 2018). Based on a laboratory comparison of different chemical

ionization techniques, Riva et al. (2019) showed that Vocus PTR-TOF is sensitive to a large range of oxygenated VOCs. With the deployment of a Vocus PTR-TOF in the French Landes forest, Li et al. (2020) observed various terpenes and terpene oxidation products, including a range of organic nitrates.

With the benefit of the capabilities of Vocus PTR-TOF to detect hundreds to thousands of molecules, a great challenge arises to analyze the complex dataset where emission sources and atmospheric physical and chemical processes are mixed

together. The characteristic analysis based solely on individually identified compounds cannot give the full picture of the measurements. Factor analytical techniques, e.g., positive matrix factorization (PMF), have been utilized to extract information

from mass spectrometer data by resolving co-varying signals with common sources or atmospheric processes into a single factor (Paatero and Tapper, 1994). For example, PMF analysis has been widely applied by the research community using aerosol mass spectrometer to identify multiple primary organic aerosol sources and SOA aging processes (Lanz et al., 2007;

Ulbrich et al., 2009; Zhang et al., 2011). Yan et al. (2016) successfully applied PMF to unit-mass-resolution (UMR) nitrate ion-based chemical ionization mass spectrometer ($NO_3^-$ CIMS) data to differentiate mainly monoterpene highly oxygenated organic molecules (HOMs) formed from different formation pathways in the boreal forest. The application of PMF to high-resolution (HR) $NO_3^-$ CIMS data by Massoli et al. (2018) identified more HOM factors at an isoprene-dominated forest site in Alabama, USA. Recently, the mass spectral binning combined with PMF (binPMF) was proposed as a novel and simple method

for analyzing high-resolution mass spectra datasets (Zhang et al., 2019a). This approach divides the full mass spectra into small bins as input data to PMF, thus avoiding the time-consuming and complicated peak identification. Zhang et al. (2019b) further applied binPMF to sub-ranges of ambient $NO_3^-$ CIMS mass spectra and separated more meaningful factors related to chemical processes yielding HOMs.

In this work, we present the first factor analysis on Vocus PTR-TOF datasets to identify and apportion the contribution

of different sources and formation pathways to atmospheric organic vapors. The measurements were conducted in two forest ecosystems in Europe, the French Landes forest and the boreal forest in southern Finland. Due to orders of magnitude differences in the signal intensities of ions between lower mass range and higher mass range, we divided the mass spectra into two sub-ranges (50–200 Th and 201–320 Th) and performed binPMF analysis on these ranges separately. While the UMR analysis loses all possible HR details and the HR peak identification introduces high uncertainties due to the complexity of

overlapping peaks, the binPMF method includes as much of the HR information as possible in a robust way. The resolved factors were linked to possible sources or chemistry processes by examining their mass profiles, time series, diurnal cycles, and correlation with molecular markers. Comparison were discussed among different factors and also between the two forests for the common sources apportioned. Based on the interpretation of the resolved factors, further insights were provided regarding the atmospheric processes of monoterpenes and sesquiterpenes.

## 2 Materials and methods

### 2.1 Site description and measurement period

The measurement data were obtained during summertime in two forest environments in Europe, the Landes forest in southwestern France and the boreal forest research station SMEAR (Station for Measuring Forest Ecosystem-Atmosphere Relations) II in southern Finland. The field campaign in the Landes forest was conducted from 8 to 20 July 2018 as part of the

Characterization of Emissions and Reactivity of Volatile Organic Compounds in the Landes forest (CERVOLAND) campaign. An overview of the Vocus PTR-TOF measurements in the Landes forest has been presented earlier by Li et al. (2020). The ambient observations at the SMEAR II station were performed during 18 June – 18 July 2019.

The Landes forest (44º29'N, 0º57'W) is the largest man-made pine forest in Europe, mainly filled with maritime pine trees (*Pinus pinaster Aiton*). The sampling site is situated at the European Integrated Carbon Observation System (ICOS) station at Bilos. The nearest urban area of the Bordeaux metropole is around 40 km to the northwest. A more detailed description of the measurement site can be found elsewhere (Kammer et al., 2018; Bsaibes et al., 2020; Li et al., 2020). Ambient meteorological parameters, including temperature, relative humidity (RH), wind speed and direction, solar radiation, and pressure, and mixing ratios of nitrogen oxides (NO and $NO_2$) and $O_3$ were continuously monitored at the station throughout the campaign.

The SMEAR II station (61º51'N, 24º17'E) is located in a boreal mixed-coniferous forest in Hyytiälä, southern Finland (Hari and Kulmala, 2005). The site is dominated by a rather homogeneous Scots pine (*Pinus sylvestris L.*) stand and represents a rural background measurement station. The nearest large city Tampere, located about 60 km to the southwest, has approximately 200 000 inhabitants. The station is equipped with extensive facilities to measure forest ecosystem-atmosphere interactions. Ambient meteorological parameters (i.e., global radiation, UVA, UVB, temperature, RH, pressure, and wind speed and direction), mixing ratios of various trace gases (i.e., carbon dioxide, carbon monoxide, sulfur dioxide, $NO_x$, and $O_3$), and particle concentration and size distribution, are continuously measured at the station.

## 2.2 Instrumentation

A Vocus PTR-TOF was deployed in both forest ecosystems to characterize atmospheric organic vapors. Equipped with a new chemical ionization source with a low-pressure reagent-ion source and focusing ion-molecule reactor (FIMR), the Vocus PTR-TOF is able to measure organic vapors with a wide range of volatilities (Krechmer et al., 2018; Riva et al., 2019; Li et al., 2020). A quadrupole radio frequency (RF) field inside the FIMR focuses ions to the central axis and improves the detection efficiency of product ions. Compared with conventional PTR instruments, the sensitivity and detection efficiency of Vocus PTR-TOF are significantly improved (detection limit < 1 pptv). With a high water mixing ratio (10% v/v–20% v/v) in the FIMR, the instrument shows no humidity dependence for sensitivity. More instrumental details have been provided elsewhere (Krechmer et al., 2018; Li et al., 2020).

During both campaigns, we operated the Vocus ionization source at a pressure of 1.5 mbar. Sample air was drawn in through a ~1-m-long PTFE tubing (10 mm o.d., 8 mm i.d.). A sample air flow of 4.5 L min$^{-1}$ was used to reduce inlet wall losses and sampling delay. Around 100–150 ccm of this flow was sampled into the Vocus and the remainder was directed to the exhaust. The mass resolving power of the long TOF mass analyzer was 12 000 – 13 000 m $\Delta$m$^{-1}$ during our measurements. Data were recorded with a time resolution of 5 s. During the campaign in the Landes forest, background checks were automatically performed every hour using ultra-high-purity nitrogen (UHP $N_2$). The instrument was calibrated twice a day using a mixture of terpenes ($\alpha$-/$\beta$-pinene+limonene; *p*-cymene). For measurements at the SMEAR II station, background measurements by injection of zero air using a built-in active carbon filter and quantitative calibrations with a multi-component standard cylinder were automatically conducted every three hours. All the *m/z* ratios mentioned in this work include the contribution from the charger ion (H$^+$, mass of 1 Th) unless stated otherwise.

## 2.3 binPMF data preparation and analysis

As described by Zhang et al. (2019a), binPMF divides the mass spectra into small bins and then takes advantage of PMF analysis to separate different sources or formation processes. The binPMF allows utilization of the high-resolution information of the complex mass spectra without the time-consuming and potentially error-prone steps of peak identification and peak fitting before the factorization. Selected peaks of interest can be analyzed after binPMF, based on the output factors. PMF assumes that factor profiles are constant and unique, and the measured signal of a chemical component is a linear combination of different factors. This approach does not require a priori information about the factors. The detailed working principle of PMF has been provided in numerous previous studies (Paatero and Tapper, 1994; Zhang et al., 2011; Yan et al., 2016).

To prepare the data and error matrices for PMF input, the Vocus PTR-TOF data were processed using the software package "Tofware" (v3.2.0; Tofwerk), which runs in the Igor Pro environment (WaveMetrics, OR, USA). The detailed data processing routines have been presented elsewhere (Stark et al., 2015). Signals were averaged over 30 min for data processing. Unlike traditional UMR or HR fitting of the mass spectra, in binPMF analysis, the mass spectra were divided into small bins after mass calibration. Due to the greater mass resolving power of the TOF mass analyzer compared with former binPMF studies (Zhang et al., 2019a, 2019b), a bin width of 0.01 Th was applied in this study. At a nominal mass $N$, signals between $N$-0.15 and $N$+0.35 Th were included for binning. The error matrix was calculated to include uncertainty from counting statistics following Poisson distribution and instrument electronic noise, as described by Yan et al. (2016) and Zhang et al. (2019a). The electric noise was estimated as the median of the standard deviation of binned noise signals between two nominal masses, with noise range between $N$+0.4 and $N$+0.6 Th.

Figure 1 shows the average mass spectra of the measurements in the Landes forest as an example. Since the signal intensity of larger molecules is generally much lower than that of low-mass molecules, we divided the mass spectra into two sub-ranges, the low mass range (51–200 Th) and the high mass range (201–320 Th). Factor analysis was separately performed on these two sub-ranges using an Igor-based open-source PMF Evaluation Tool (PET; http://cires1.colorado.edu/jimenez-group/wiki/index.php/PMF-AMS_Analysis_Guide#PMF_Evaluation_Tool_Software). We ran the PMF up to ten factors for both sub-ranges. For the low mass range of 51–200 Th, the signals at $m/z$ 81 Th ($C_6H_8H^+$, monoterpene fragment) and 137 Th ($C_{10}H_{16}H^+$, monoterpenes) were markedly higher than the others. Because PMF assumes that the data matrix can be explained by a linear combination of different factors, even a very tiny fraction of these high peaks is split into a factor, they may dominate the mass profile of the factor. As shown in Fig. S1, with the inclusion of $C_6H_8H^+$ and $C_{10}H_{16}H^+$, the mass profiles of several factors were quite similar and dominated by these peaks. Therefore, the major mass bins of these ions were excluded for further PMF analysis, but their corresponding isotopes were retained, effectively downweighting their contributions to the PMF result. This simple approach by removing the main peaks of the largest signals produced factors that made sense both chemically and through their temporal behaviors, lending confidence in the results. However, to quantify the relative contribution of different factors, the signals of these removed mass bins were counted back into their corresponding factors. More details can be found in Sect. 4.4.

**2.4 Estimation of OH and NO₃ radicals**

The OH concentration was calculated by scaling the measured UVB radiation intensity with the empirically derived factors
from Petäjä et al. (2009) and Kontkanen et al. (2016):

$$[OH]_{proxy} = \left( \frac{8.4 \times 10^{-7}}{8.6 \times 10^{-10}} UVB^{0.32} \right)^{1.92}$$

Measurements of NO₃ concentration is challenging. The concentration of NO₃ radical was calculated by assuming a steady state between its production from $O_3$ and $NO_2$ and its removal by oxidation reactions and losses in the atmosphere. Details can be found in Allan et al. (2000) and Peräkylä et al. (2014).

**3 Dataset overview**

Figure 2 shows the temporal behaviors of temperature, global radiation, concentrations of $O_3$ and $NO_x$, and concentrations of isoprene and monoterpenes in the Landes forest and at the SMEAR II station. In the Landes forest, the weather was mainly sunny during the observation period (global radiation > 400 W m⁻²), indicating strong photochemical activity. The air mass in the forest was largely influenced by local sources, with wind speeds below canopy lower than 1 m s⁻¹ over the whole campaign. The $O_3$ concentration fluctuated dramatically between day and night, with the average daytime concentration peaking up to 50 ppb and the average nighttime level falling below 2 ppb (Li et al., 2020). The low $O_3$ concentration at night was probably to some extent caused by its titration by monoterpenes (Fig. 2a; Kammer et al., 2018, 2020). The Landes forest is known for strong monoterpene emissions (Simon et al., 1994). During our measurements, the average mixing ratios of isoprene and monoterpenes were 0.6 ppb and 6.0 ppb, respectively. More details about this dataset can be found in Li et al. (2020). All data in the Landes forest are reported in local time and all data at the SMEAR II station in Finnish winter time (both equal UTC time + 2).

During the measurements at the SMEAR II station, 84% (26 out of 31) of the days had strong photochemistry (global radiation > 400 W m⁻²), with the rest being cloudy days. The diurnal variation in $O_3$ concentration was not as dramatic as that in the Landes forest. In the daytime, the $O_3$ concentration sometimes reached up to 50 ppb. At night, the $O_3$ level still largely remained high, above 20 ppb, in contrast to the observations in the Landes forest. A possible explanation is less nighttime $O_3$ consumption by terpenes at the SMEAR II station. On average, the mixing ratios of isoprene and monoterpenes were 0.2 ppb and 0.8 ppb, respectively, during the measurements, much lower than those in the Landes forest.

## 4 Results and discussion

### 4.1 Choice of PMF solution and factor interpretation

To interpret the PMF results, the most critical decision is to choose the best number of factors. More factors introduce more degrees of freedom to explain variations in the data, but too many factors may cause splitting of real factors and lead to mathematical artifacts without physical meaning (Ulbrich et al., 2009). The factor interpretation results in this work are summarized in Table 1. In the factor name, L means the Landes forest and S means the SMEAR II station.

For the low mass range of the Landes forest dataset, the $Q/Q_{exp}$ varied from 15.5 to 6.0 for two to ten factors ($Q$ is the

195 total sum of the squares of the scaled residuals for PMF solutions). The larger $Q/Q_{exp}$ indicates underestimation of the errors or high residuals for some bins that cannot be simply modeled by the solution (Ulbrich et al., 2009). After seven factors, increasing the factor number does not significantly decrease the $Q/Q_{exp}$ (step change < 7%). The optimal solution of seven factors was chosen by evaluating the variations of $Q/Q_{exp}$ vs. varying factor number, the distribution of the scaled residuals for each $m/z$, sum of the squares of scaled residuals, factor mass profile, factor time series and diurnal cycles, and also signs of

200 split factors (Ulbrich et al., 2009; Zhang et al., 2011). Figure S2 shows the distribution of scaled residuals as a function of $m/z$. For some bins the residuals are still high (the scaled residuals as high as ±200). The seven factors include Factor L1 closely related to the $C_4H_8H^+$ ion, Factor L2 attributed to a plume event occurring on a single night during the campaign, Factor L3 mainly containing lightly oxidized compounds with six or seven carbon atoms ("$C_6$" or "$C_7$"), Factor L4 representing monoterpenes, Factor L5 indicative of isoprene and its oxidation products, Factor L6 identified as unknown source with large

contributions from unknown peaks, and Factor L7 dominated by monoterpene lightly oxidized compounds. The direct comparison of the mass spectra, time series, and diurnal cycles of six-factor and eight-factor solutions are shown in Fig. S3 and Fig. S4. In the six-factor case, the $C_4H_8H^+$ ion-related factor cannot be separated. With eight-factor results, the factor representing isoprene and its oxidation products is split into two components with similar time series. For the high mass range of the Landes forest dataset, the $Q/Q_{exp}$ decreased from 2.5 to 0.9 for two to ten factors. After evaluation, we choose the eight-

factor solution to explain the data. The $Q/Q_{exp}$ value of the eight-factor solution was 1.1 and the decreasing trend in $Q/Q_{exp}$ obviously slowed down after eight factors. The distribution of scaled residuals as a function of $m/z$ for the eight-factor solution is shown in Fig. S5. The eight factors are interpreted as Factor L8 dominated by lightly oxygenated compounds containing 13 carbon atoms ("$C_{13}$"), Factor L9 attributed to a plume event occurring on a single night during the campaign, Factor L10 mainly related to sesquiterpene lightly oxidized compounds, Factor L11 representing more oxidized products mainly from

monoterpene oxidation, Factor L12 indicating sesquiterpenes, Factor L13 largely composed of monoterpene-derived organic nitrates, Factor L14 mainly containing oxidized compounds with twelve, fourteen or sixteen carbon atoms ("$C_{12}$", "$C_{14}$" or "$C_{16}$")and Factor L15 as unknown source largely contributed by siloxane compounds. Figure S6 and Figure S7 display the mass spectra, time series, and daily variations of seven-factor and nine-factor solutions. In the seven-factor case, monoterpene more oxidized products and monoterpene-derived organic nitrates are mixed together into a single factor. However, in the

nine-factor solution, the unknown factor mainly composed of siloxane compounds is split into two factors with similar mass profiles and similar diurnal trends.

For the SMEAR II dataset, the optimal solutions of five-factor and four-factor are chosen for the low and high mass ranges, respectively. The $Q/Q_{exp}$ varied from 7.2 to 2.5 for two to ten factors in the low mass range and from 2.0 to 1.0 for two to ten factors in the high mass range. The five factors for the low mass range are identified as Factor S1 - $C_4H_8H^+$ ion-related,

Factor S2 - monoterpenes, Factor S3 - lightly oxidized compounds with six to nine carbon atoms, Factor S4 - isoprene and its oxidation products, and Factor S5 - monoterpene lightly oxidized compounds. The mass spectra, time series, and diurnal profiles of the four-factor and six-factor solutions for the low mass range are presented in Fig. S8 and Fig. S9. For the four-factor solution, monoterpene lightly oxidized products are not separated as a single factor and mixed into the others. In the six-factor case, the factor indicative of monoterpene lightly oxidized products is split into two factors. The four factors for the high

mass range include Factor S6 - sesquiterpene lightly oxidized products, Factor S7 - sesquiterpenes, Factor S8 - more oxidized compounds, and Factor S9 - unknown source. The direct comparison of the mass spectra, time series, and diurnal variations of three-factor and five-factor solutions are shown in Fig. S10 and Fig. S11. The three-factor solution does not identify a factor representing sesquiterpenes. In the five-factor case, the factor of unknown source mainly contributed by siloxane compounds is split into two factors with similar mass profiles.

The rotational freedom of the PMF solutions was explored through the use of the FPEAK parameters. For each of the optimal solutions, we varied the FPEAK values between -1 and +1 with the step of 0.2. For the low mass ranges of the Landes and SMEAR II dataset, the varying FPEAK values did not change the factor profiles and time series much, indicating that varying FPEAK values from -1 to +1 did not affect the overall results of PMF analysis. For the high mass range of the Landes measurements, we saw variations especially in the factor profiles by varying FPEAK values. But after a detailed evaluation,

we found no evidence that solutions with FPEAK values away from zero were preferable. However, for the high mass range of the SMEAR II measurements, the solutions with positive values of FPEAK worked better than that with FPEAK = 0 in terms of factor profiles. The factor time series were similar when FPEAK values varied. But for the factor profiles with positive FPEAK values, the factor of monoterpene more oxidized products including organic nitrates contained less traces of siloxanes and showed elevated fractions of the corresponding fingerprint peaks (Fig. S12). As discussed in Sect. 4.3, these siloxanes can

come from cosmetics and personal care products, and silicone oils used in instrument pumps. The temporal variations of these siloxanes differed significantly from those of monoterpene more oxidized products. After evaluation, we chose the solution with FPEAK = +0.6 for the high mass range of the SMEAR II dataset, where siloxanes feature more in one factor.

**4.2 Source identification in the Landes forest**

Figure 3 and Figure 4 illustrate the factor profiles, time series, and diurnal variations of the seven factors resolved in the low

mass range. For the high mass range, the mass spectra of the five factors are shown in Fig. 5, and their time series and daily variations in Fig. 6. The high-resolution peak fitting was further performed on the mass profile to identify the fingerprint peaks

in the factors. Fingerprint peaks are defined largely based on their distribution in the factors rather than their absolute intensity in the mass profile. The correlation map of each factor with various compounds is shown in Fig. S13.

### 4.2.1 Low mass range (51–200 Th)

*Factor L1: $C_4H_8H^+$ ion-related*

Factor L1 shows irregular diurnal variations with spiky peaks in the time series (Fig. 4b). The major bins that are largely distributed into this factor are $C_4H_8H^+$ and $C_4H_{10}O_2H^+$. Factor L1 closely correlates with these fingerprint peaks. Considering the high signal intensity of $C_4H_8H^+$ ion and its large contribution to this factor, we name Factor L1 as $C_4H_8H^+$ ion-related.

*Factor L2: A plume event*

Factor L2 is identified as a plume event occurring on a single night during the campaign. As shown in Fig. 4a, the time series of this factor are characterized by much higher intensities at midnight of 9 July 2018 than over the other days. Fingerprint peaks in this factor are aromatic compounds such as $C_6H_6H^+$, $C_7H_6H^+$, and $C_6H_6OH^+$. Factor L2 is well correlated with benzene and phenol ($r^2 = 0.88$; Fig. S13).

*Factor L3: $C_6$ and $C_7$ lightly oxidized products*

The diurnal cycle of Factor L3 exhibits a small morning peak at 9:00 and significantly elevated intensities during nighttime, peaking at around 22:00 (Fig. 4b). As illustrated in the mass profile of Factor L3, this factor is mainly composed of lightly oxidized compounds containing six or seven carbon atoms such as $C_6H_{10}OH^+$, $C_7H_{10}OH^+$, $C_6H_{10}O_2H^+$, and $C_7H_{12}O_2H^+$.

*Factor L4: monoterpenes*

The mass profile of Factor L4 is dramatically characterized by a monoterpene peak ($^{13}CC_9H_{16}H^+$) and its major fragments 270 (i.e., $^{13}CC_5H_8H^+$ and $C_7H_8H^+$) inside the instrument. As shown in Fig. 4b, the diurnal variation of this factor follows a similar pattern to that of monoterpenes (Li et al., 2020). The signal intensity of the factor starts to increase at 20:00, peaks at midnight, and then drops to around the detection limit during daytime. Monoterpene emissions are mainly influenced by temperature (Hakola et al., 2006; Kaser et al., 2013). Therefore, with the continuous emissions of monoterpenes and the shallow boundary layer at night, the signal intensities of monoterpenes are observed to be elevated. The signal of $C_{10}H_{16}OH^+$ is also mostly 275 resolved into this factor. $C_{10}H_{16}O$ could be primary emissions of oxygenated monoterpenes or monoterpene oxidation products (Kallio et al., 2006; McKinney et al., 2011). Previous ambient observation has demonstrated that the atmospheric behavior of $C_{10}H_{16}O$ has high similarity to that of monoterpenes (Li et al., 2020).

*Factor L5: isoprene and its oxidation products*

The marker peaks in Factor L5 are highly dominated by isoprene and its major oxidation products in the atmosphere, i.e., 280 $C_5H_8H^+$ and $C_4H_6OH^+$ (Wennberg et al., 2018). Isoprene emissions strongly depend on light intensity (Monson and Fall, 1989; Kaser et al., 2013) and generally show high concentrations in the day. Similarly, the daily variations of Factor L5 display maximum signal during daytime and minima at night.

*Factor L6: unknown source*

Factor L6 is characterized by increased signals in the afternoon. The major peaks in its factor profile are $C_6H_4O_2H^+$,
$C_6H_6O_3H^+$, and numerous unidentified peaks with negative mass defect. As this factor is clearly separated as a single source with high signals during our observations and the molecule markers remain unidentified, we name this factor as an unknown source.

*Factor L7: monoterpene lightly oxidized products*

Fingerprint peaks in this factor are monoterpene oxidation products with oxygen number from one to three, such as $C_9H_{14}OH^+$, $C_{10}H_{14}OH^+$, $C_{10}H_{16}O_2H^+$, and $C_{10}H_{16}O_3H^+$. This factor displays clear morning and evening peaks, similar to the behavior of these lightly oxidized compounds (Li et al., 2020).

### 4.2.2 High mass range (201–320 Th)

*Factor L8: $C_{13}$ lightly oxidized products*

The mass profile of Factor L8 is characterized by high peaks of lightly oxidized compounds containing 13 carbon atoms, like $C_{13}H_{18}O_2H^+$ and $C_{13}H_{20}O_3H^+$. Similar to $C_6$ and $C_7$ lightly oxidized compounds, this factor shows a morning peak at 9:00 and an evening peak at around midnight (Fig. 6b).

*Factor L9: A plume event*

Factor L9 is characterized with much higher intensities on a single night (9 July 2018) during the campaign (Fig. 6a). Fingerprint peaks in the mass profile of Factor L9 are numerous unidentified ions. The time series of Factor L9 correlate tightly with aromatic compounds $C_6H_6$ and $C_6H_6O$ ($r^2 = 0.75$).

*Factor L10: sesquiterpene lightly oxidized products*

The fingerprint peaks identified in this factor are $C_{15}H_{22}OH^+$, $C_{15}H_{24}OH^+$, $C_{15}H_{22}O_2H^+$, $C_{15}H_{24}O_2H^+$, and $C_{15}H_{24}O_3H^+$, which are typical reaction products from sesquiterpene oxidation (Fu et al., 2009; Yee et al., 2018). The signal intensity of this factor is generally high during nighttime, but shows another morning peak at 8:00. In addition to the production from sesquiterpene oxidation processes, $C_{15}H_{22}O$ and $C_{15}H_{24}O$ can be oxygenated sesquiterpene alcohols and aldehydes directly emitted from vegetation (Kännaste et al., 2014).

*Factor L11: monoterpene more oxidized products*

The mass spectrum of this factor is mainly characterized by more oxidized compounds from monoterpene oxidation such as $C_{10}H_{16}O_4H^+$, $C_{10}H_{14}O_5H^+$, $C_{10}H_{16}O_5H^+$, and $C_{10}H_{16}O_6H^+$. As shown in Fig. S13, the time series of Factor L11 show good correlations with these compounds. Compared with monoterpene lightly oxidized compounds, the diurnal cycle of this factor shows a broad daytime distribution peaking between 14:00 and 20:00, caused by strong and complex photochemical reactions during the day.

*Factor L12: sesquiterpenes*

The mass spectra of Factor L12 are clearly dominated by a big single peak of $C_{15}H_{24}H^+$, indicating the influence of sesquiterpenes. Sesquiterpene emissions from plants are found to exhibit a strong dependence on temperature (Duhl et al., 2008). Therefore, similar to the diurnal cycle of Factor L4, this factor shows prominently enhanced signals during nighttime.

*Factor L13: monoterpene-derived organic nitrates*

The signal intensity of this factor starts to increase in the early morning (around 7:00) and presents a distinct morning peak at 9:00. In addition, a much smaller evening peak is observed at 21:00. The daily variations of this factor are quite similar to those of monoterpene-derived organic nitrates measured in the Landes forest (Li et al., 2020). Consistently, the major peaks in the factor profile are $C_{10}H_{15}NO_4H^+$, $C_{10}H_{15}NO_5H^+$, $C_9H_{13}NO_6H^+$, and $C_{10}H_{15}NO_6H^+$, indicating the dominant contribution of organic nitrates formed from monoterpene oxidation processes.

*Factor L14: $C_{12}$, $C_{14}$ or $C_{16}$ lightly oxidized compounds*

The mass profile of Factor L14 is characterized with distinct peaks of $C_{12}$, $C_{14}$ or $C_{16}$ lightly oxidized compounds, i.e., $C_{12}H_{26}O_3H^+$, $C_{14}H_{26}O_2H^+$, $C_{16}H_{30}O_2H^+$, and $C_{16}H_{30}O_3H^+$. The time series of Factor L14 correlate very well with those of $C_{12}H_{26}O_3$ ($r^2 = 0.83$), characterized with enhanced signals during daytime and low intensities at night (Fig. 6b). $C_{12}H_{26}O_3$ has been found during the photooxidation of dodecane (Zhang et al., 2014).

*Factor L15: unknown source*

The mass profile of Factor L15 is predominantly characterized by high cyclic volatile methyl siloxanes (VMSs) peaks and some unidentified peaks (Fig. 5). The major cyclic VMSs are protonated D3 siloxane, D4 siloxane, and their $H_3O^+$ cluster ions, which have been widely used in cosmetics and personal care products (Buser et al., 2013; Yucuis et al., 2013). The diurnal cycle of this factor shows a bit higher intensity during daytime but also big background signals at night. A similar factor has also been identified at the SMEAR II station. More detailed discussions can be found in Sect. 4.3.2.

## 4.3 Source identification in the southern Finnish boreal forest

The factor profiles, time series, and diurnal cycles of the five-factor solution for the low mass range are presented in Fig. 7 and Fig. 8. Figure 9 and Figure 10 present the mass spectra, time series, and daily variations of the four factors identified in the higher mass range at the SMEAR II station. The correlation coefficients among each factor and various fingerprint compounds can be found in Fig. S14.

### 4.3.1 Low mass range (51–200 Th)

*Factor S1: $C_4H_8H^+$ ion-related*

Similar to the source identification in the Landes forest, a factor related to $C_4H_8H^+$ ion is clearly resolved at the SMEAR II station. The major peaks in this factor are $C_4H_8H^+$, $C_4H_{12}O_2H^+$, and $C_4H_{14}O_3H^+$.

*Factor S2: monoterpenes*

A factor representing monoterpenes is also identified at the SMEAR II station, with fingerprint peaks of $^{13}CC_5H_8H^+$, $C_7H_{10}H^+$, and $^{13}CC_9H_{16}H^+$. Monoterpenes undergo some degree of fragmentation within PTR instruments, and $C_6H_8H^+$ and $C_7H_{10}H^+$ have been observed to be the major fragments of monoterpenes (Tani et al., 2003; Tani, 2013; Kari et al., 2018). The signal intensity of monoterpenes at the SMEAR II station is much lower than that in the Landes forest.

*Factor S3: $C_6$-$C_9$ lightly oxygenated compounds*

The mass profile of Factor S3 is characterized by lightly oxygenated compounds with carbon atoms varying from six to nine ($C_6$-$C_9$) such as $C_6H_{10}OH^+$, $C_6H_{12}OH^+$, $C_7H_{10}OH^+$, $C_8H_{14}OH^+$, and $C_9H_{18}OH^+$. The signal intensity of this factor shows high peaks at night and low appearance during daytime. These lightly oxygenated molecules can be directly emitted from anthropogenic and biogenic sources or come from oxidation processes of various VOC precursors (Conley et al., 2005; Pandya et al., 2006; Rantala et al., 2015; Hartikainen et al., 2018). For instance, $C_7H_{10}O$ has been found from direct soil emissions (Abis et al., 2020) or oxidation processes of 1,2,4-trimethyl benzene (Mehra et al., 2020). Therefore, we expect the molecules in this factor to be either directly emitted or as oxidation products of forest emissions.

*Factor S4: isoprene and its oxidation products*

At the SMEAR II station, a factor largely composed of isoprene and its oxidation products is also resolved. The outstanding peaks in the factor profile are $C_5H_8H^+$, $C_4H_6OH^+$, $C_4H_8O_2H^+$, and $C_5H_8O_2H^+$. The signal intensity of this factor is around ten times lower than that of Factor L5 measured in the Landes forest. Similar to previous isoprene observations at the sampling site (Hakola et al., 2012), this factor shows a broad daytime peak and low signals at night.

*Factor S5: monoterpene lightly oxidized products*

Similar to Factor L7 identified in the Landes forest, this factor is characterized by major peaks of monoterpene lightly oxidized compounds, as shown in Fig. 7. The signal intensity of this factor starts to increase at 20:00 and presents an obvious morning peak at 7:00.

**4.3.2 High mass range (201-320 Th)**

*Factor S6: sesquiterpene lightly oxidized products*

This factor is identified as sesquiterpene lightly oxidized compounds with high peaks of $C_{14}H_{22}OH^+$, $C_{14}H_{24}OH^+$, $C_{15}H_{22}OH^+$, and $C_{15}H_{24}OH^+$, similar to Factor L10 in the Landes forest. The time series of this factor show strong correlations with the lightly oxidized products of sesquiterpenes (Fig. S14; $r^2 > 0.88$).

*Factor S7: sesquiterpenes*

Similar to Factor L12 in the Landes forest, this factor is characterized by the big peak of $C_{15}H_{24}H^+$, demonstrating the dominance of sesquiterpenes in the factor. Figure S14 shows that this factor closely correlates with monoterpenes and sesquiterpenes, with $r^2$ being 0.73 and 0.85, respectively. Compared with the identification of Factor L12, representing sesquiterpenes in the Landes forest, the signal intensity of this factor at the SMEAR II station is approximately three times lower. Including the lower signals of monoterpenes and isoprene, the results indicate weaker biogenic VOC emissions in the Hyytiälä boreal forest than in the Landes forest.

*Factor S8: monoterpene more oxidized products including organic nitrates*

Factor S8 is mainly composed of more oxidized compounds, particularly from monoterpene oxidation processes, including monoterpene-derived organic nitrates. The major peaks are shown in Fig. 9. Mixed with monoterpene-derived organic nitrates, this factor of more oxidized compounds displays a small morning peak at 8:00 and generally high signals during daytime (Fig. 10).

*Factor S9: unknown source*

The marker peaks of Factor S9 are mainly high cyclic volatile methyl siloxanes (VMSs) and unidentified compounds (Fig. 9), i.e., protonated D3 siloxane, D4 siloxane, and their $H_3O^+$ cluster ions. In addition to cosmetics and personal care products, siloxanes can also be emitted by silicone oils (Schweigkofler et al., 1999), which have been widely used in instrument pumps (Gonvers et al., 1985). In this study, the temporal behaviors of Factor S9 are contributed by high background signals and present a very regular diurnal cycle with higher signal intensities during daytime and lower ones at night, which basically follow the variations in ambient temperature. Therefore, we speculate that Factor S9 is mainly caused by emissions from silicone oil pumps used by several instruments in the container, and these emissions are influenced by daily temperature changes.

## 4.4 Comparison among different factors

The monoterpene factor and sesquiterpene factor correlate very well with each other at both sites (Fig. 11; $r^2 = 0.69$ in the Landes forest and $r^2 = 0.59$ at the SMEAR II station). The emissions of monoterpenes and sesquiterpenes are both strongly influenced by temperature. Their signals peak at night with the effect of the shallow boundary layer. In the daytime, the low signals of the monoterpene and sesquiterpene factors are likely a combination of enhanced atmospheric mixing after sunrise and the rapid photochemical consumption of monoterpenes and sesquiterpenes. The signal of monoterpene factor is around 15 times higher than that of sesquiterpene factor at the SMEAR II station while it is around 60 times in the Landes forest. Previous studies found that sesquiterpene emissions from pines, spruces, and birches under normal conditions were 5-15% of total monoterpene emissions by mass (Rinne et al., 2009 and references therein), in line with our observations.

In the Landes forest, a factor of $C_6$ and $C_7$ lightly oxidized products (Factor L3) was resolved in the low mass range and a factor representative of $C_{13}$ lightly oxidized products (Factor L7) was identified in the high mass range. Interestingly, these two factors show a close correlation with each other ($r^2 = 0.64$). The $C_6$ oxygenated compounds have been observed during the oxidation processes of benzene and $C_7$ oxygenated compounds from toluene oxidation (Sato et al., 2012; Zaytsev et al., 2019). These compounds can also be directly emitted from biogenic or anthropogenic sources (Conley et al., 2005; Pandya et al., 2006; Rantala et al., 2015). The temporal behaviour of Factor L7 is similar to that of Factor L3, indicating potentially similar formation pathways of these lightly oxygenated compounds. Therefore, the $C_{13}$ oxidized compounds are speculated to be produced through the dimer formation mechanisms of $C_6$ and $C_7$ species (Valiev et al., 2019). In addition, $C_{13}H_{20}O_3$ can be direct emissions of methyl jasmonate (Meja), which is a typical green leaf volatile used in plant-plant communications for defensive purposes (Cheong and Choi, 2003). But considering the close correlation between Factor L3 and Factor L7, we conclude that these C13 lightly oxidized compounds are formed from atmospheric oxidation processes, not direct plant emissions.

Monoterpene lightly oxidized products and sesquiterpene lightly oxidized products were resolved as individual factors at both sites (Factor L7 vs. Factor L10 in the Landes forest and Factor S5 vs. Factor S6 at the SMEAR II station). While the diurnal variations of monoterpene lightly oxidized products are similar to those of sesquiterpene lightly oxidized products,

their time series do not follow very well with each other, suggesting the probably different formation pathways or different factors influencing the atmospheric processes of monoterpenes and sesquiterpenes. More discussions can be found in Sect. 4.6.

In this study, the source apportionment analysis was performed separately on two subranges of the mass spectra. It can happen that the same factor is identified in both subranges. For example, both Factor L2 and Factor L9 are defined as the

plume event during the measurements. The time series of Factor L2 and Factor L9 show a high correlation coefficient of 0.93 and correlate tightly with aromatic compounds, indicating the major influence of anthropogenic sources. As mentioned above, the air masses in the Landes forest were relatively stable during our observations with wind speed below canopy < 1 m s$^{-1}$. Therefore, the influence of long-range regional transport on the atmosphere in the forest is expected to be minor. We speculate that the plume event is a result of local anthropogenic disturbances favored by the lower boundary layer height at night.

**4.5 Comparison between the two forests**

To give an overview of the source distributions in the two forest ecosystems, we calculated the mass fraction of each factor based on their average signal intensities. We acknowledge that it is not a perfect method to quantify the contributions of various sources and formation processes. The sensitivities of different VOCs measured by the PTR instruments may vary by a factor of 2-3 (Sekimoto et al., 2017; Yuan et al., 2017). The uncertainties can come from the challenge to convert the signal intensity

to atmospheric concentrations because of problematic calibrations, especially given that many unknown molecules exist in the mass spectra. The major bins at $m/z$ 81 Th and 137 Th, which were initially excluded to perform PMF analysis, were counted into their corresponding factors. For example, the signals of the discarded bins at $m/z$ 81 Th and 137 Th were estimated by multiplying their isotope signals by the corresponding scale number and added to the factor representing monoterpenes. The average mass fractions of various PMF factors in total measured organic vapors are shown in Fig. 12.

While the atmospheric environment and ecosystem processes differ markedly in the Landes forest and the southern Finnish boreal forest, the results of this study reveal similar biogenic sources and oxidation processes in these forest environments. For instance, the biogenic VOCs at the two sites are both dominated by monoterpenes, with the average fractions of 29% in the Landes forest and at the SMEAR II station. These two forests are both characterized by pine trees, with dominant emissions of $\alpha$-pinene and $\beta$-pinene (Riba et al., 1987; Simon et al., 1994; Hellén et al., 2018). According to the PMF results,

isoprene and its major oxidation products in these environments (mainly $C_4H_6O$) contribute 14% and 21% in the two ecosystems, respectively. Factors indicative of sesquiterpenes are identified in the high mass range at both sites. The average contribution of sesquiterpenes (0.5% in the Landes forest and 1.7% at the SMEAR II station) is much smaller than that of monoterpenes and isoprene. Factors of the lightly oxidized products, more oxidized products, and organic nitrates of monoterpenes/sesquiterpenes in total contribute 8% and 12% of the measured organic vapors in the Landes forest and at the

SMEAR II station, respectively.

The factor related to $C_4H_8H^+$ ion was resolved at both sites and contributes 10% in the Landes forest and 16% at the SMEAR II station. According to the discussions by Li et al. (2020), the observation of $C_4H_8H^+$ in the Landes forest can be

attributed to several sources. For instance, the protonated butene may contribute to the $C_4H_8H^+$ signal, which is emitted by biogenic or anthropogenic sources (Hellén et al., 2006; Zhu et al., 2017). Another possible explanation is that the $C_4H_8H^+$ ion
is produced during the fragmentation of many VOCs in the PTR instruments (Pagonis et al., 2019). The green leaf volatiles (GLV) have been found to fragment at $m/z$ 57 Th inside the PTR instruments, which are a group of six-carbon aldehyde, alcohols and their esters released by plants. Furthermore, butanol can easily lose an OH during the PTR source ionization and produce prominent $C_4H_8H^+$ peaks (Spanel and Smith, 1997). Therefore, the condensation particle counters (CPCs) using butanol for aerosol measurements at the site could also be an important source of $C_4H_8H^+$ ions, although the exhaust air from
these instruments has been filtered using charcoal denuder. At the SMEAR II station, the bivariate polar plot where the concentrations of air pollutants are shown as a function of WS and WD indicates that high signals of $C_4H_8H^+$ generally occur when the wind comes from the north (Fig. S15). Located in the north of the measurement container is a particle measurement cottage with several CPCs inside using butanol. A previous study at this station also found that $C_4H_8H^+$ signals detected by PTR-TOF mainly come from butanol used by aerosol instruments (Schallhart et al., 2018). Therefore, it is expected that Factor
S1 at the SMEAR II station is mainly contributed by butanol fragmentation inside the instrument where butanol comes from nearby aerosol instruments.

Figure 13 presents the comparison of the mass spectra of the common sources identified at both sites, with the $x$ and $y$ axis showing the mass fraction of different bins in the factor profile. The scattering in the plots is mainly caused by mass bins with much lower mass fractions. However, the dominant bins with high mass contributions in the factor profiles generally
correlate well and are located close to the 1:1 line. It shows that the mass spectra of the common sources match well in these two forests and the sources and processes are indeed similar despite the quite different regions the forests are in.

## 4.6 Insights into terpene oxidation processes

Terpenes undergo varying degrees of oxidation in the atmosphere and produce a large variety of organic compounds with different volatilities (Donahue et al., 2012; Ehn et al., 2014). With the sub-range PMF analysis performed in this study, terpene
reaction products with varying oxidation degrees are successfully separated. The sources of monoterpene lightly oxidized products, sesquiterpene lightly oxidized products, monoterpene more oxidized compounds, and monoterpene-derived organic nitrates are identified in both forests with distinct characteristics. These factors account for 8-12% of the measured organic vapors in the two forests. It provides a great opportunity to gain insights into terpene oxidation processes. Because some environmental parameters, for example, measurements of UVB to estimate OH concentration, are not available in the Landes
forest, the results from SMEAR II station are presented as follows.

### 4.6.1 Monoterpene oxidation

The oxidation processes of monoterpenes at the SMEAR II station have been investigated by several previous studies, mostly based on the highly oxidized compounds. Utilizing non-negative matrix factorization analysis on iodide-adduct CIMS data at the SMEAR II station, Lee et al. (2018) found that the gas-phase organic species subgroup of $C_{6-10}H_yO_{\geq 7}$ showed distinct

daytime diel trends. Yan et al. (2016) conducted source apportionment of HOMs at the SMEAR II station and separated various HOM formation pathways, such as monoterpene ozonolysis and monoterpene oxidation initiated by $NO_3$ radical. In this study, three types of monoterpene reaction products were detected: monoterpene lightly oxidized compounds, monoterpene more oxidized compounds, and monoterpene-derived organic nitrates. The latter two were not clearly separated into different factors at the SMEAR II station due to the similarities in their overall time trends. For example, the time series of $C_{10}H_{15}NO_5H^+$

correlate well with those of $C_{10}H_{16}O_4H^+$ and $C_{10}H_{16}O_5H^+$ ($r^2 > 0.61$).

Consistent with previous observations, monoterpene more oxidized products (i.e., $C_{10}H_{16}O_4$ and $C_{10}H_{14}O_5$) have a broad high distribution throughout the day due to the active photochemical processes during daytime. Monoterpene-derived organic nitrates (i.e., $C_{10}H_{17}NO_4$, $C_{10}H_{15}NO_5$, and $C_9H_{13}NO_6$) are mainly characterized by a distinct morning peak at around 8:00, approximately 2 h after the NO peak. But their intensities are also elevated at night. PMF analysis of $NO_3^-$ CIMS dataset

observed similar diurnal variations of terpene organic nitrates factor at a forest site in the southeastern US (Massoli et al., 2018). Compared with β-pinene and most other monoterpenes, the overall organic nitrate yield from α-pinene + $NO_3$ is rather low (Fry et al., 2014; Kurtén et al., 2017). Laboratory studies found that using iodide-adduct FIGAERO-HR-ToF CIMS, $C_{10}H_{15}NO_6$ is the most abundant organic nitrate in both gas- and particle-phase measurements of α-pinene + $NO_3$ reactions (Nah et al., 2016). Boyd et al. (2015) mainly detected $C_{10}H_{17}NO_4$, $C_{10}H_{15}NO_5$, $C_{10}H_{17}NO_5$, and $C_{10}H_{15}NO_6$ with iodide-adduct

CIMS from the α-pinene + $NO_3$ system. Using $C_{10}H_{17}NO_5$ and $C_{10}H_{15}NO_6$ as the examples, we checked their correlations with the products of [OH] × [monoterpenes], [$O_3$] × [monoterpenes], and [$NO_3$] × [monoterpenes] in different periods of the day (Fig. 14; Fig. S16). Comparatively, $C_{10}H_{17}NO_5$ and $C_{10}H_{15}NO_6$ correlate better with the products of [OH] × [monoterpenes] and [$O_3$] × [monoterpenes] during daytime (9:00~18:00). However, for the product of [$NO_3$] × [monoterpenes], its correlation coefficients with $C_{10}H_{17}NO_5$ and $C_{10}H_{15}NO_6$ are a bit higher at night (20:00 to 4:00 of the next day). These results indicate that

monoterpene-derived organic nitrates can be mainly formed by the $NO_3$-initiated oxidation at night, but in daytime by the OH and $O_3$-initiated oxidation followed by NO termination of the $RO_2$. It should be noted that $C_{10}H_{17}NO_5$ and $C_{10}H_{15}NO_6$ are used as examples because both of them are fingerprint peaks of the factor, but in real environments it may not be the case that these molecules are always produced from the above formation routes.

**4.6.2 Sesquiterpene oxidation**

The lightly oxygenated compounds from sesquiterpene reactions present a big morning peak and elevated signal intensities at night, similar to the diurnal variations of monoterpene lightly oxidized products. Hellén et al. (2018) showed that at the SMEAR II station, $O_3$ oxidation dominated the first step of sesquiterpene reactions for the whole year. It has also been observed in central Amazonia that sesquiterpenes contributed the highest to total $O_3$ reactivity although sesquiterpene concentrations were much lower than those of monoterpenes and isoprene (Yee et al., 2018). At the SMEAR II station, emissions of

sesquiterpenes are dominated by β-caryophyllene (Hellén et al., 2018). Photooxidation of β-caryophyllene in the chamber experiments resulted in high aerosol yield and is expected to strongly influence SOA formation (Jaoui et al., 2013). Using the mass spectrometric techniques, Jokinen et al. (2016) observed the production of highly oxidized organic compounds from β-

caryophyllene ozonolysis, i.e., monomers $C_{15}H_{24}O_{7,9,11}$ and $C_{15}H_{22}O_{9,11}$, and dimers $C_{29}H_{46}O_{12,14,16}$ and $C_{30}H_{46}O_{12,14,16}$. However, due to the instrumental limitation, only the lightly oxidized products from sesquiterpene reactions were identified in this study.

Interestingly, a strong RH-dependence was observed for the correlations between sesquiterpene lightly oxidized compounds and the product of [OH] $\times$ [sesquiterpenes] or [O$_3$] $\times$ [sesquiterpenes]. These products represent the oxidation rates of sesquiterpenes with OH radical and O$_3$. As shown in Fig. 15, the corresponding correlation coefficients vary significantly with RH. In addition, the signal intensities of sesquiterpene lightly oxidized products also show high dependence on RH. At lower RH (RH<40%), the signal intensities of sesquiterpene lightly oxidized products are relatively low and correlate closely with the product of [OH] $\times$ [sesquiterpenes] and [O$_3$] $\times$ [sesquiterpenes]. The high signal intensities of sesquiterpene lightly oxidized products occur when RH>70% but the correlation between sesquiterpene lightly oxidized compounds and the product of [OH] $\times$ [sesquiterpenes] or [O$_3$] $\times$ [sesquiterpenes] is more scattered. Such high RH-dependence was not observed for monoterpene lightly oxidized compounds (Fig. S17). These findings have not been observed by previous studies and the reasons behind remain unclear. High-RH conditions typically occur during nights with temperature inversion (Zha et al., 2018), while RH below 40% generally only occurs at the station during sunny days. The controlling role of temperature can be ruled out because temperature is strongly anti-correlated with RH and is known to influence terpene emissions and terpene reaction rates. Future studies are needed to dig deep into the atmospheric processes of sesquiterpenes and monoterpenes.

## 5 Concluding remarks

In this study, we conducted Vocus PTR-TOF measurements in two forest environments and performed binPMF analysis on these complex mass spectra. In addition to VOC species, Vocus PTR-TOF is able to measure large amounts of oxygenated VOCs with enhanced detection efficiency. According to the results in this work, factor analysis on Vocus PTR-TOF mass spectra separated VOC precursors and their reaction products with varying oxidation degrees into different factors. These factors showed distinct characteristics in the atmosphere. Comparatively, the conventional PTR instruments or gas chromatograph-mass spectrometry (GC-MS) largely detect VOC precursors of low-mass molecules (Dewulf et al., 2002; de Gouw et al., 2007). Previous source apportionment studies on these datasets mainly identified primary biogenic and anthropogenic emission sources (Vlasenko et al., 2009; Patokoski et al., 2014; Baudic et al., 2016; Debevec et al., 2017; Sarkar et al., 2017; Wang et al., 2020). Recently, factorization methods have been applied on NO$_3^-$ CIMS dataset to identify various atmospheric formation pathways of HOMs (Yan et al., 2016; Massoli et al., 2018; Zhang et al., 2019b). Here, for the first time, source apportionment of Vocus PTR-TOF data identified various primary emission sources and secondary formation pathways of atmospheric organic vapors, highlighting the novelty of Vocus PTR-TOF in measuring both VOCs and oxygenated VOCs. The relative abundances of organic precursors, the lightly oxidized products, and the more oxidized products can be utilized by modellers to evaluate simulation output, improve model performance, and provide new perspectives to understand gas-phase physicochemical processes.

Compared with VOC species, VOC reaction products are generally present in much smaller amounts in the atmosphere. Therefore, utilizing a sub-range PMF analysis, or other similarly weighting method, is particularly important for Vocus PTR-TOF observations, where several orders of magnitude differences are expected between VOC precursors and their oxidation products. Compared with the low mass range, the average contributions of the high mass range in total signals are significantly smaller, 2% and 9%, in the Landes forest and at the SMEAR II station, respectively. However, the identified factors in the high mass range, such as sesquiterpenes, sesquiterpene lightly oxidized products, monoterpene-derived organic nitrates, and more oxidized compounds, can provide crucial insights into atmospheric physicochemical processes. For example, we found that the correlations between sesquiterpene lightly oxidized compounds and the products of [OH] × [sesquiterpenes] or [$O_3$] × [sesquiterpenes] show strong dependences on RH. High signal intensities of sesquiterpene lightly oxidized compounds only occur at high-RH conditions. Such high RH-dependence was not observed for monoterpene lightly oxidized compounds.

To summarize, this study successfully performed binPMF analysis on sub-ranges of mass spectrometry dataset acquired with a Vocus PTR-TOF in two European forest ecosystems, the Landes forest and a southern Finnish boreal forest. Both primary emission sources and secondary oxidation processes of organic vapors were identified in the two environments, particularly for terpenes and their reaction products with varying oxidation degrees (including organic nitrates). Factors of the lightly oxidized products, more oxidized products, and organic nitrates of monoterpenes/sesquiterpenes accounted for 8-12% of the measured gas-phase organic vapors in the two forests. Further interpretations show a strong RH-dependence for the behaviour of sesquiterpene lightly oxidized products but not for that of monoterpene lightly oxidized products, for which the reasons behind need more investigations in the future.

*Data Availability.* The time series of the measured trace gases, meteorological parameters, and the concentrations of isoprene and monoterpenes in the Landes forest and at the SMEAR II station are available from https://doi.org/10.5281/zenodo.3946644 (Li, 2020).

*Author contributions.* HL and ME conceived the study. HL, MR, ST, LH, PMF, EV, and EP conducted the field measurements. HL carried out the data analysis. MC, YZ, ME, and FB participated in the discussions on data analysis. HL wrote the paper with inputs from all coauthors.

*Competing interests.* Manjula R. Canagaratna and Douglas Worsnop both work for Aerodyne Research Inc.

*Acknowledgements.* This work was supported by the H2020 European Research Council (grant nos. ATM-GP (742206), COALA (638703), and CHAPAs (850614)) and the Academy of Finland (grant nos. 317380 and 320094). We thank the SMEAR II station staff for their help during field measurements in Hyytiälä. The authors also would like to thank the PRIME-QUAL program for financial support (ADEME, convention#1662C0024) and the French National Research Agency (ANR)

in the frame of the "Investments for the Future" program, within the Cluster of Excellence COTE (ANR-10-LABX-45) of the University of Bordeaux for financial support. Harald Stark and Donna T. Sueper from Aerodyne Research Inc. are acknowledged for helpful discussions.

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

Table 1. Summary of source identification results for the two forest sites (L, Landes; S, SMEAR II).

| | Factor name | Possible source/chemistry | Fingerprint molecules |
|---|---|---|---|
| **Landes forest** | | | |
| | Factor L1 | $C_4H_8H^+$ ion-related | $C_4H_8H^+$, $C_4H_{10}O_2H^+$ |
| | Factor L2 | A plume event | $C_6H_6H^+$, $C_7H_6H^+$, $C_6H_6OH^+$, unidentified peaks |
| | Factor L3 | $C_6$ and $C_7$ lightly oxidized products | $C_6H_{10}OH^+$, $C_7H_{10}OH^+$, $C_6H_{10}O_2H^+$, $C_7H_{12}O_2H^+$, |
| | Factor L4 | Monoterpenes | $C_6H_8H^+$, $C_7H_{10}H^+$, $C_{10}H_{16}H^+$ |
| | Factor L5 | Isoprene and its oxidation products | $C_5H_8H^+$, $C_4H_6OH^+$, $C_4H_6O_3H^+$ |
| | Factor L6 | Unknown source | $C_6H_4O_2H^+$, $C_6H_6O_3H^+$, unidentified peaks with negative mass defect |
| | Factor L7 | Monoterpene lightly oxidized products | $C_9H_{14}OH^+$, $C_{10}H_{14}OH^+$, $C_{10}H_{16}O_2H^+$, $C_{10}H_{16}O_3H^+$ |
| | Factor L8 | $C_{13}$ lightly oxidized products | $C_{13}H_{18}O_2H^+$, $C_{13}H_{20}O_3H^+$ |
| | Factor L9 | A plume event | Unidentified peaks |
| | Factor L10 | Sesquiterpene lightly oxidized products | $C_{15}H_{22}OH^+$, $C_{15}H_{24}OH^+$, $C_{15}H_{22}O_2H^+$, $C_{15}H_{24}O_2H^+$, $C_{15}H_{24}O_3H^+$ |
| | Factor L11 | Monoterpene more oxidized products | $C_{10}H_{16}O_4H^+$, $C_{10}H_{14}O_5H^+$, $C_{10}H_{16}O_5H^+$, $C_{10}H_{16}O_6H^+$ |
| | Factor L12 | Sesquiterpenes | $C_{15}H_{24}H^+$ |
| | Factor L13 | Monoterpene-derived organic nitrates | $C_{10}H_{15}NO_4H^+$,    $C_{10}H_{15}NO_5H^+$,    $C_9H_{13}NO_6H^+$, $C_{10}H_{15}NO_6H^+$ |
| | Factor L14 | $C_{12}$, $C_{14}$ or $C_{16}$ lightly oxidized products | $C_{12}H_{26}O_3H^+$, $C_{14}H_{26}O_2H^+$, $C_{16}H_{30}O_2H^+$ |
| | Factor L15 | Unknown source | D3 siloxane, D4 siloxane, unidentified peaks |
| **SMEAR II** | | | |
| | Factor S1 | $C_4H_8H^+$ ion-related | $C_4H_8H^+$, $C_4H_{12}O_2H^+$, $C_4H_{14}O_3H^+$ |
| | Factor S2 | Monoterpenes | $C_6H_8H^+$, $C_7H_{10}H^+$, $C_{10}H_{16}H^+$ |
| | Factor S3 | $C_6$-$C_9$ lightly oxygenated compounds | $C_6H_{10}OH^+$, $C_6H_{12}OH^+$, $C_7H_{10}OH^+$, $C_8H_{14}OH^+$, $C_9H_{12}O_2H^+$ |
| | Factor S4 | Isoprene and its oxidation products | $C_5H_8H^+$, $C_4H_6OH^+$ |
| | Factor S5 | Monoterpene lightly oxidized products | $C_9H_{14}OH^+$, $C_{10}H_{14}OH^+$, $C_{10}H_{16}OH^+$, $C_{10}H_{16}O_2H^+$, $C_{10}H_{16}O_3H^+$ |
| | Factor S6 | Sesquiterpene lightly oxidized products | $C_{14}H_{22}OH^+$, $C_{14}H_{24}OH^+$, $C_{15}H_{22}OH^+$, $C_{15}H_{24}OH^+$ |
| | Factor S7 | Sesquiterpenes | $C_{15}H_{24}H^+$ |
| | Factor S8 | Monoterpene more oxidized products including organic nitrates | $C_{10}H_{16}O_4H^+$, $C_{14}H_{22}O_3H^+$, $C_{15}H_{24}O_3H^+$, $C_{10}H_{17}NO_3H^+$, $C_9H_{13}NO_6H^+$ |
| | Factor S9 | Unknown source | D3 siloxane, D4 siloxane, unidentified peaks |

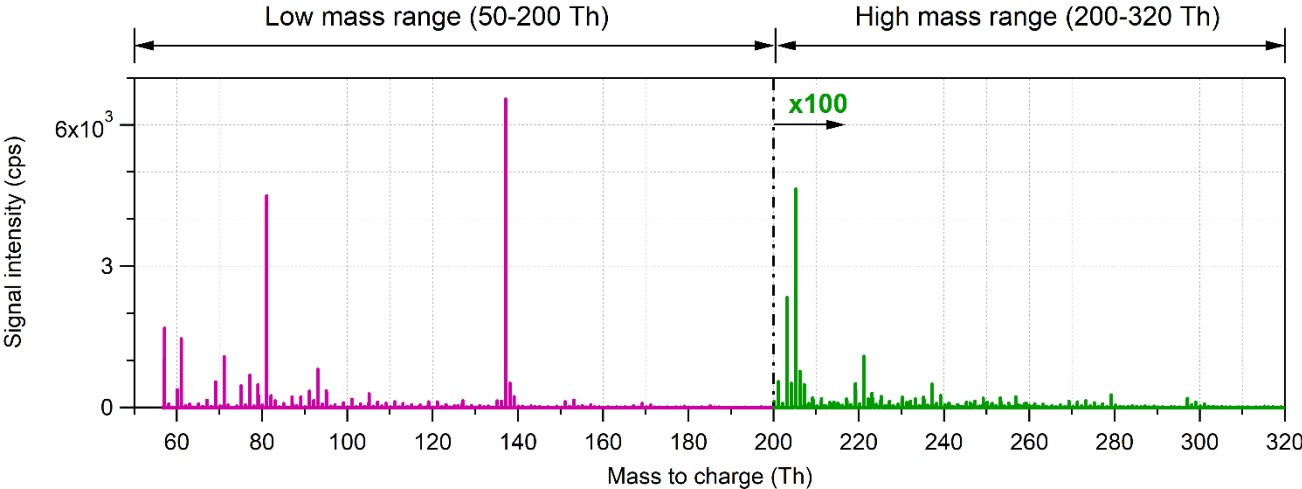

**Figure 1. Average mass spectrum measured in the Landes forest. The mass spectrum is divided into two sub-ranges for further source identification analysis. The intensity scale is shown 100-fold for the high mass range (201-320 Th).**

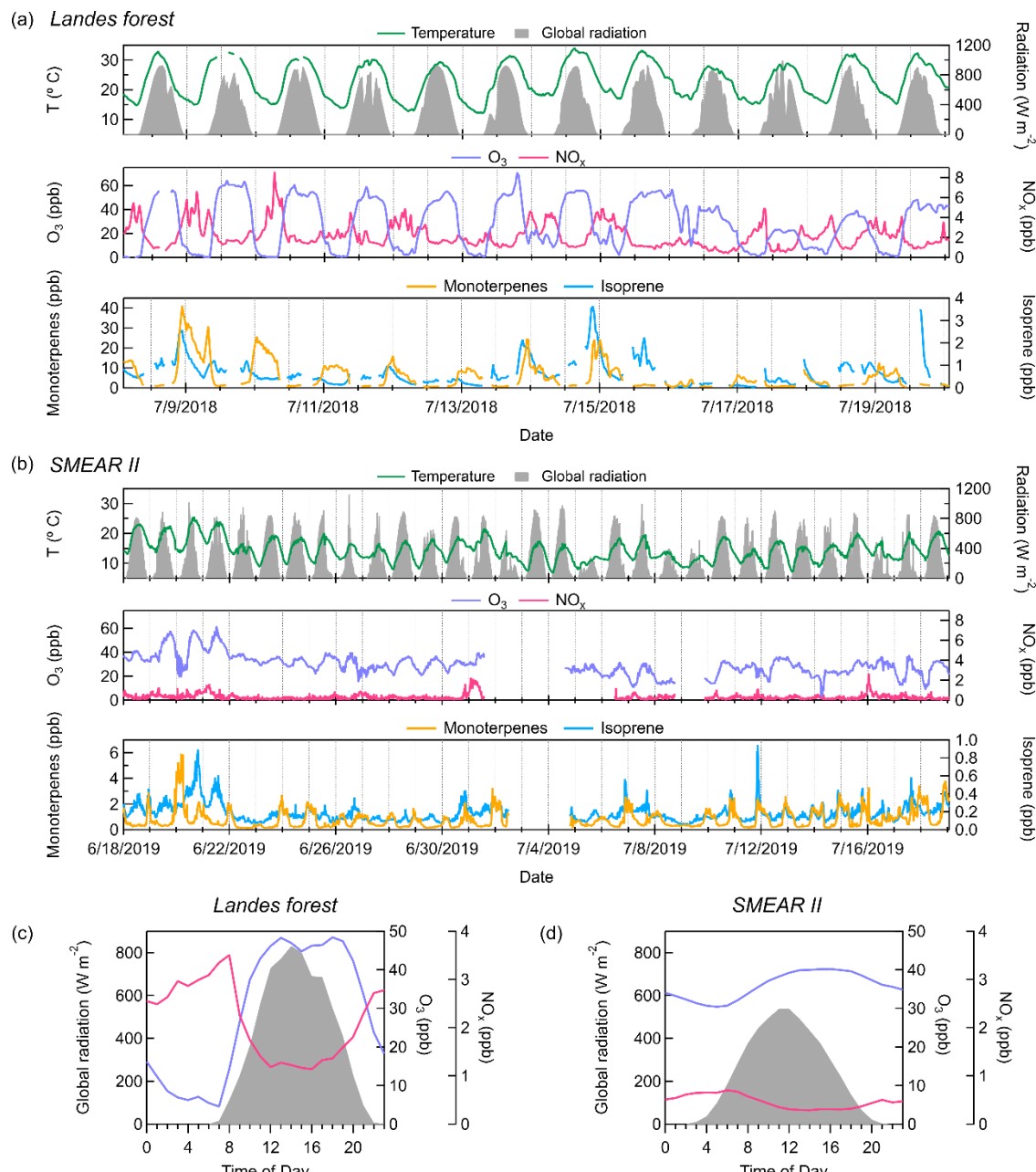

**Figure 2. Time series of temperature, global radiation, concentrations of O₃ and NOₓ, and mixing ratios of monoterpenes and isoprene throughout the measurements (a) in the Landes forest and (b) at the SMEAR II station. Average diurnal cycles of global radiation, O₃ concentration (in blue), and NOₓ concentration (in pink) (c) in the Landes forest and (d) at the SMEAR II station. All parameters, except monoterpenes and isoprene, are shown in the same y-axis scale for the two sites. Monoterpene and isoprene concentrations are much lower at the SMEAR II station than in the Landes forest.**

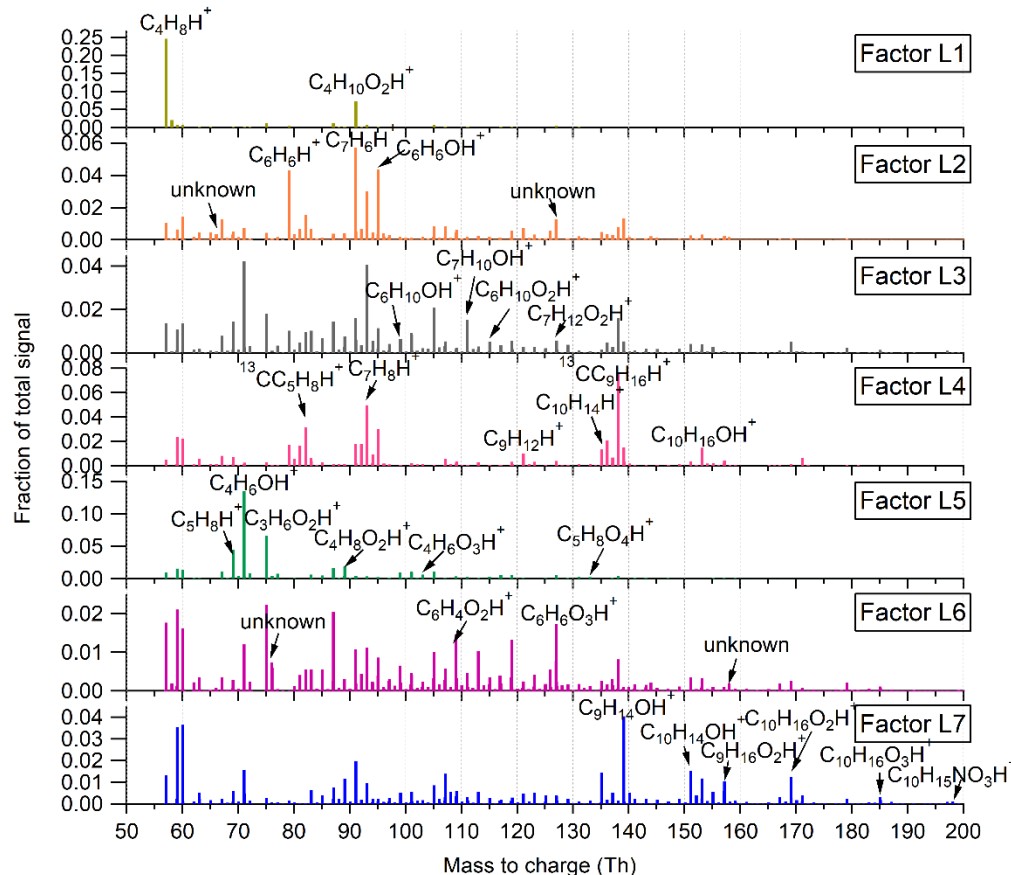

**Figure 3. Mass profiles of the seven factors resolved in the low mass range in the Landes forest. Fingerprint peaks identified by high-resolution peak fitting are shown in the mass spectra.**

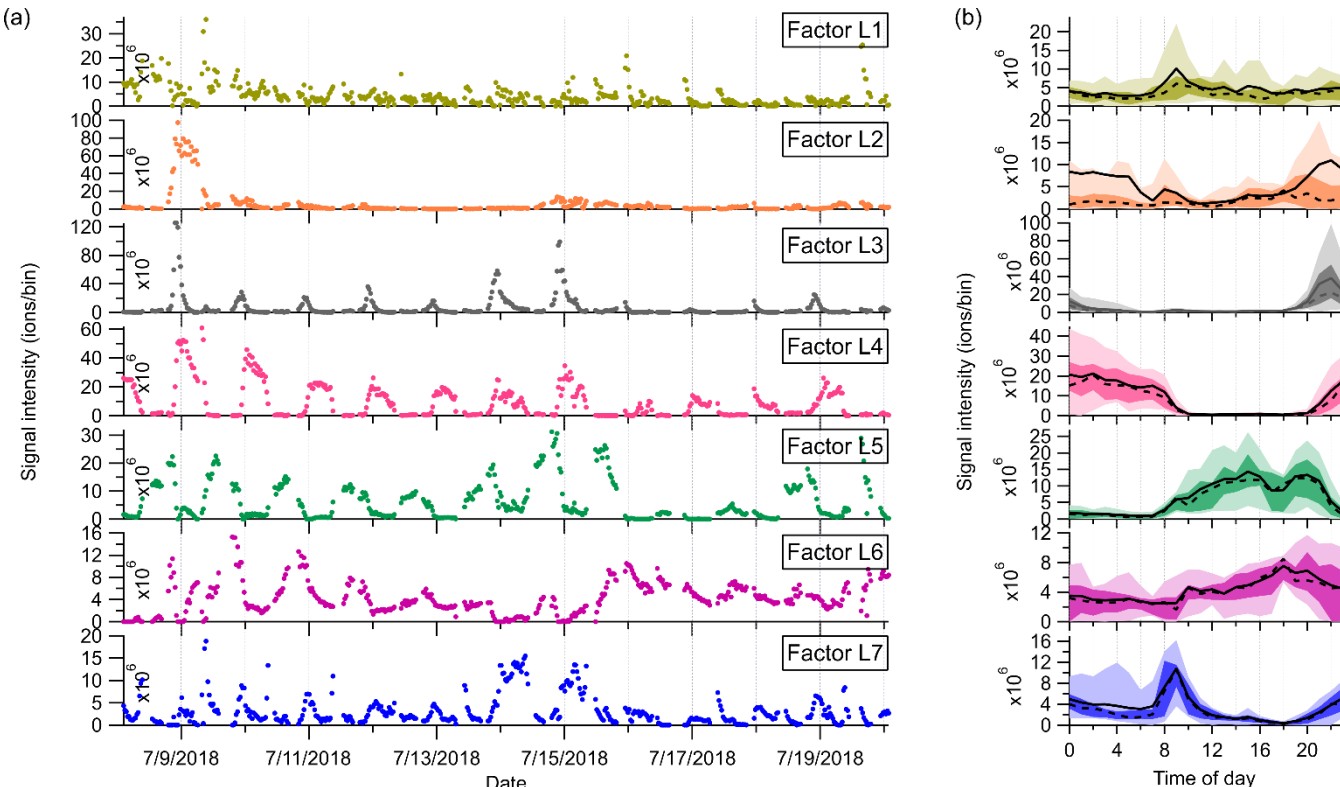

**Figure 4. (a)** Time series and **(b)** diurnal variations of the seven factors identified in the low mass range in the Landes forest. The solid and dashed lines in the diurnal plots show the mean and median values, respectively, and the shaded area shows 10th, 25th, 75th, and 90th percentiles.

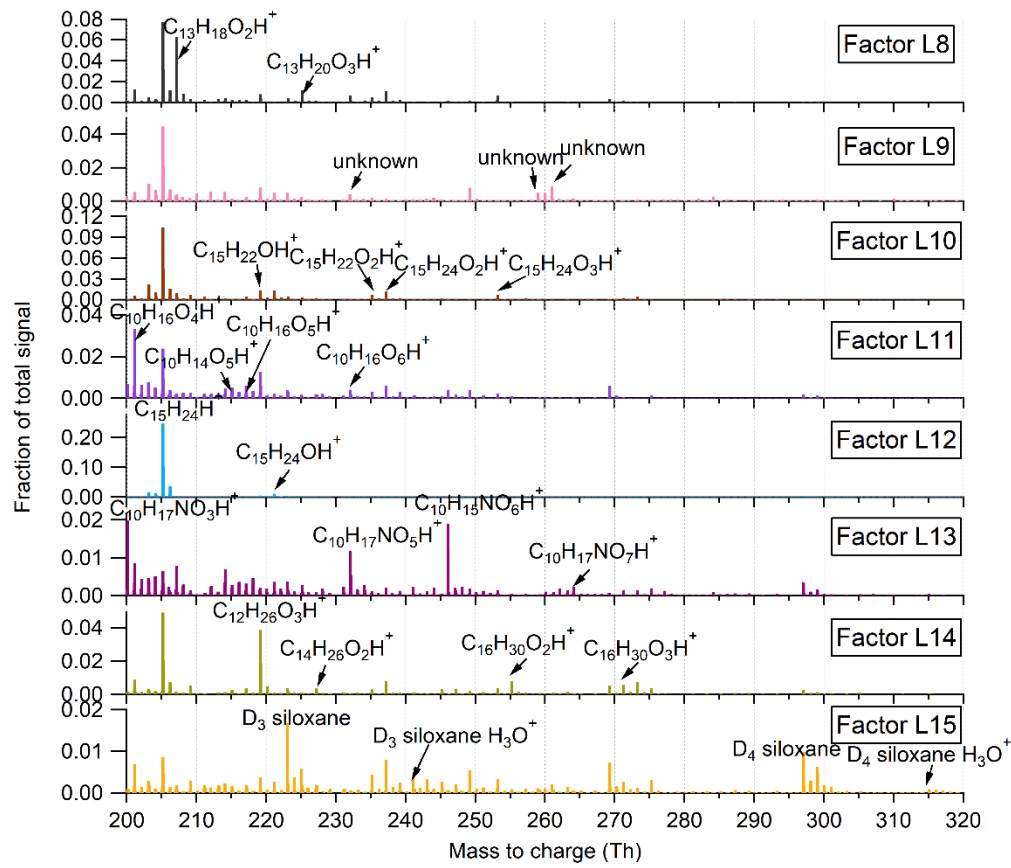

**Figure 5. Mass profiles of the eight factors resolved in the high mass range in the Landes forest, with major fingerprint peaks labeled in the mass spectra.**

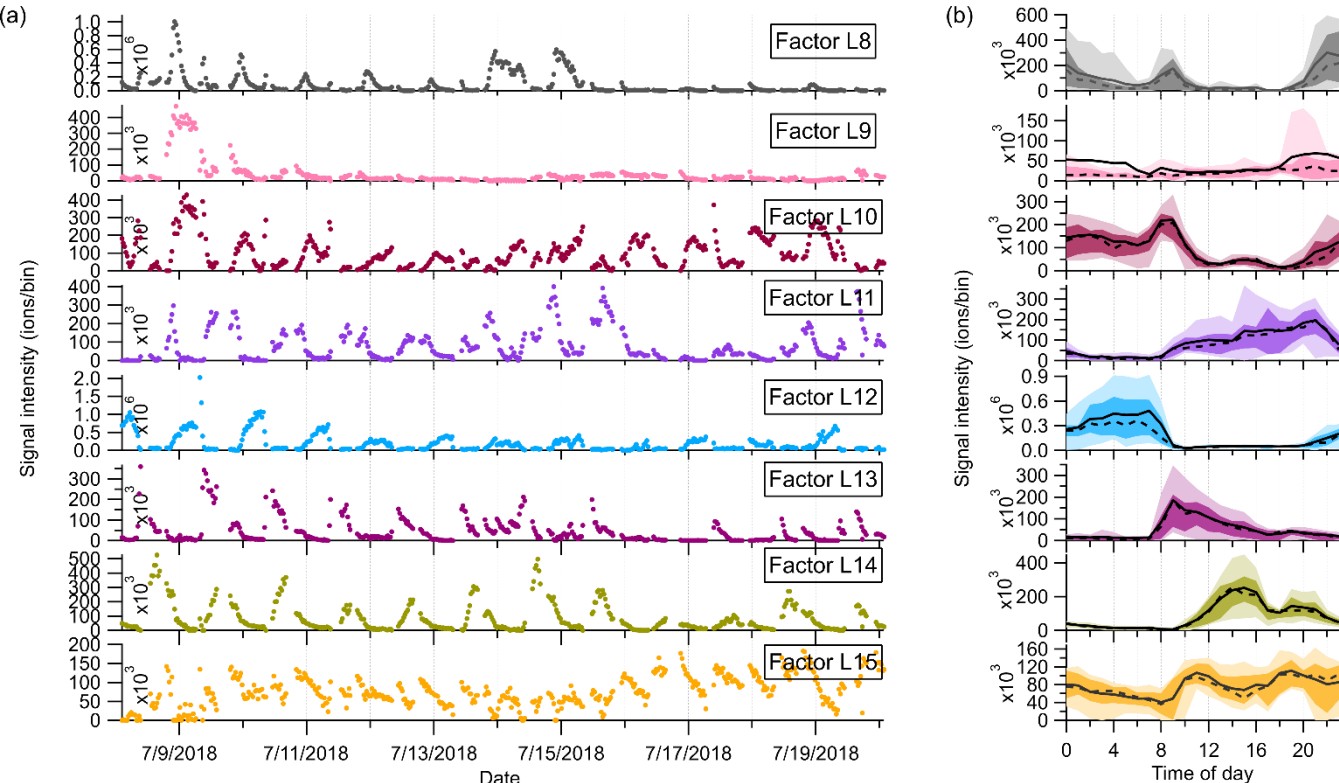

Figure 6. (a) Time series and (b) diurnal trends of the eight factors resolved in the high mass range in the Landes forest. The solid and dashed lines in the diurnal plots show the mean and median values, respectively, and the shaded area shows 10th, 25th, 75th, and 90th percentiles.

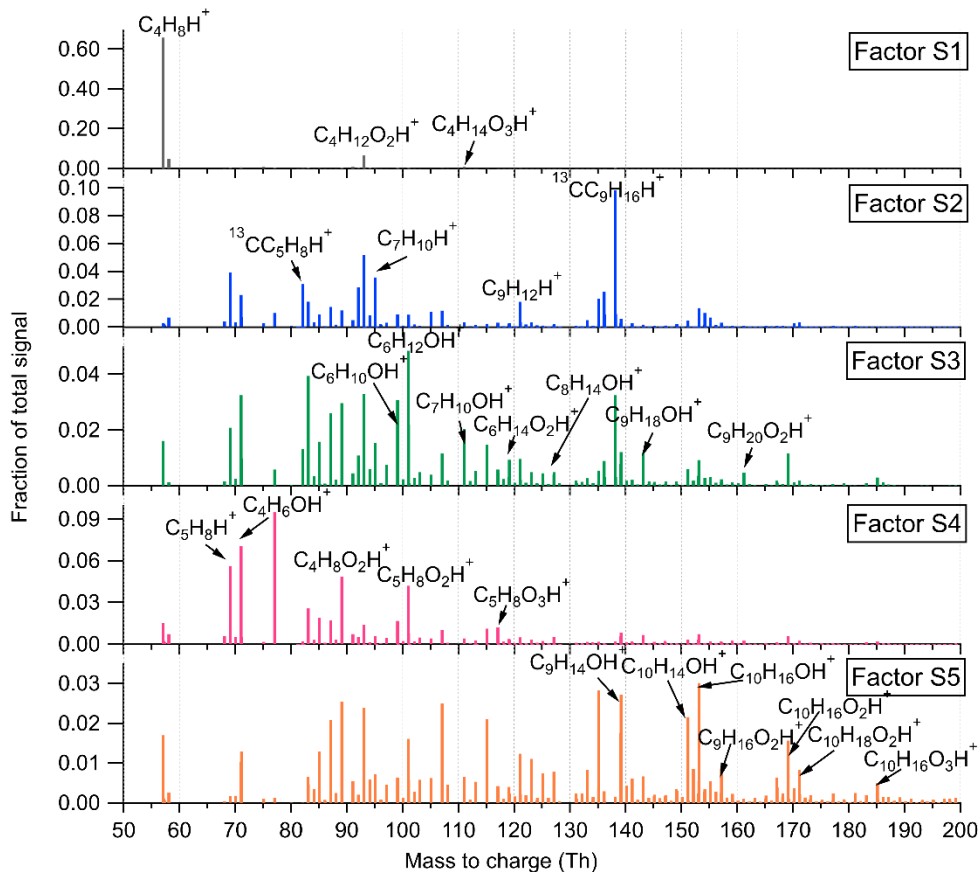

Figure 7. Mass profiles of the five factors identified in the low mass range at the SMEAR II station, with fingerprint peaks shown in the mass spectra.

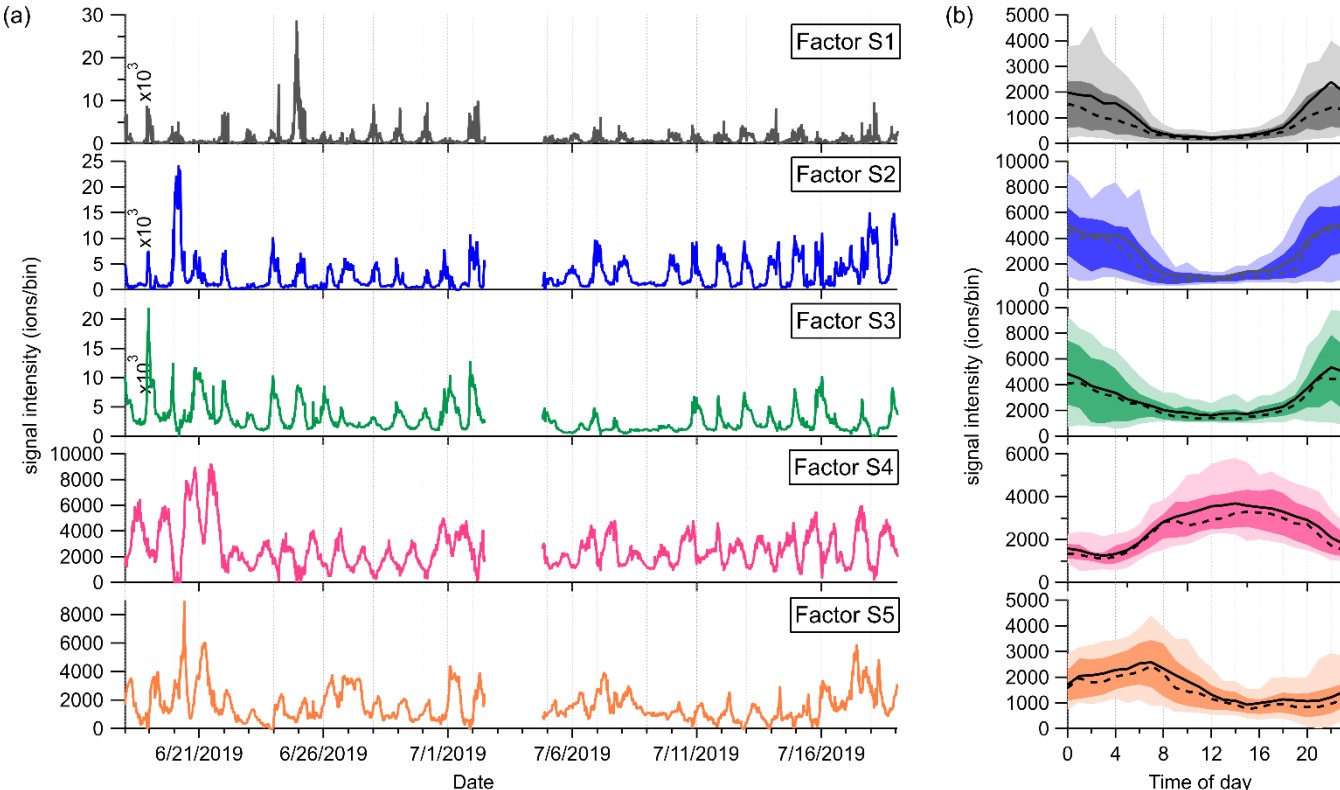

**Figure 8. (a) Time series and (b) diurnal cycles of the five factors in the low mass range at the SMEAR II station. The solid and dashed lines in the diurnal plots show the mean and median values, respectively, and the shaded area shows 10th, 25th, 75th, and 90th percentiles.**

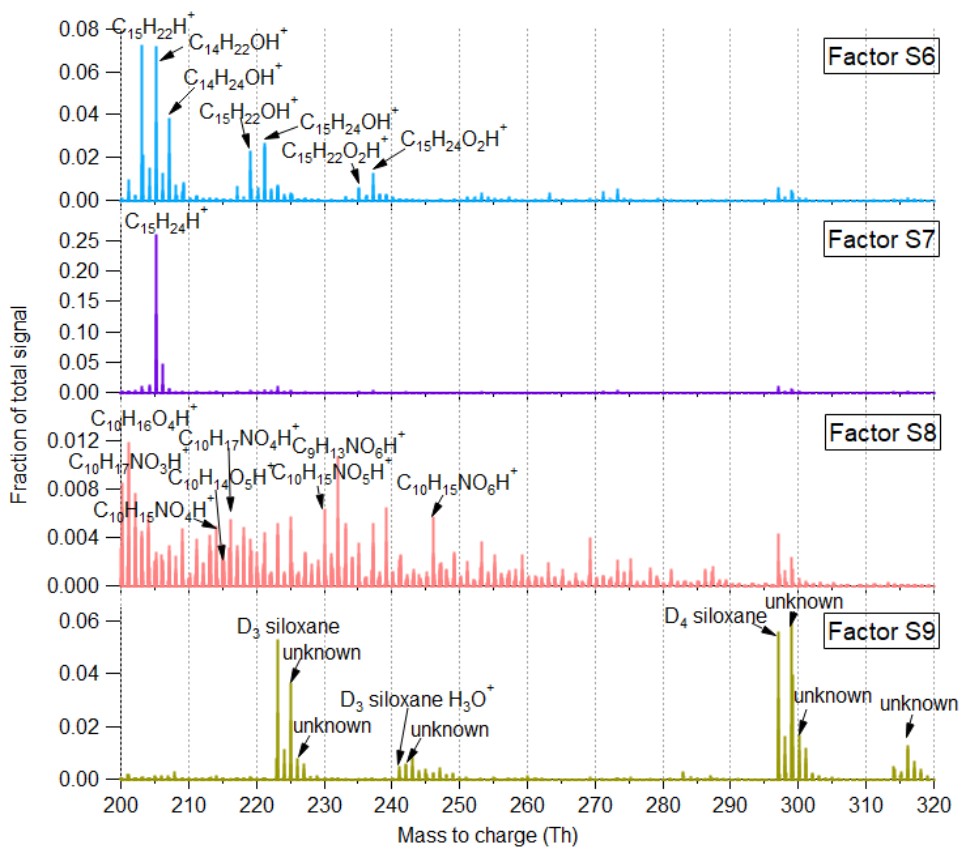

960

**Figure 9. Mass profiles of the four factors resolved in the high mass range at the SMEAR II station. The fingerprint peaks are labeled in the mass spectra.**

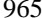

**Figure 10. (a) Time series and (b) daily trends of the four factors in the high mass range at the SMEAR II station. The solid and dashed lines in the diurnal plots show the mean and median values, respectively, and the shaded area shows 10th, 25th, 75th, and 90th percentiles.**

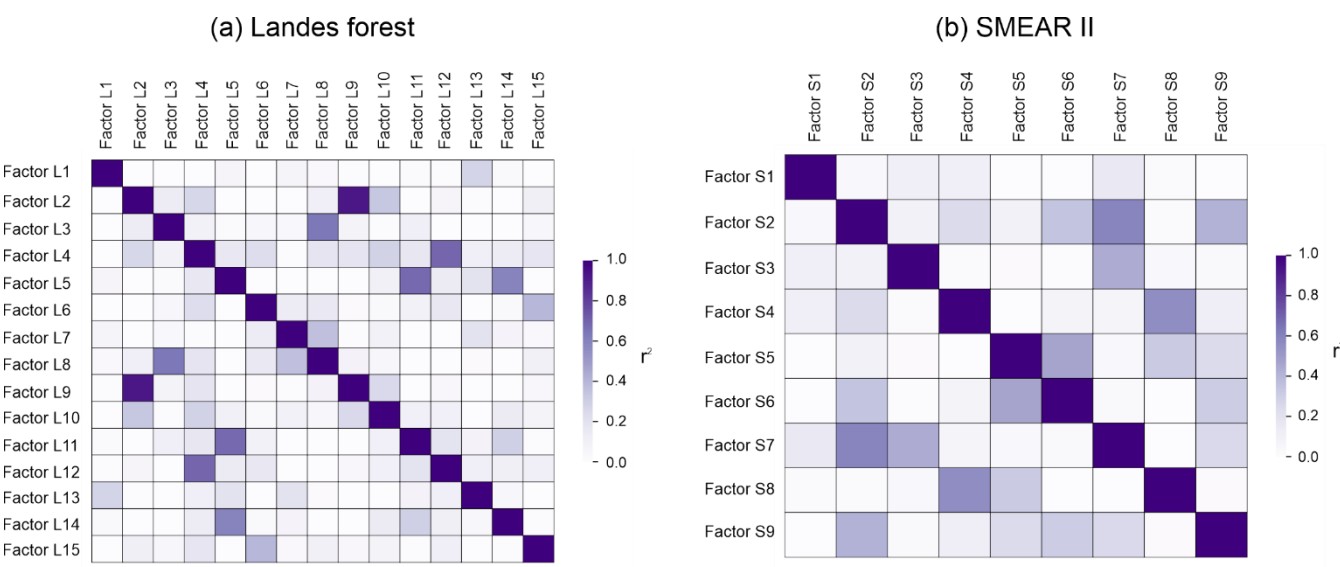

**Figure 11. The correlations among various factors identified (a) in the Landes forest and (b) at the SMEAR II station, with the color representing the correlation coefficients ($r^2$).**

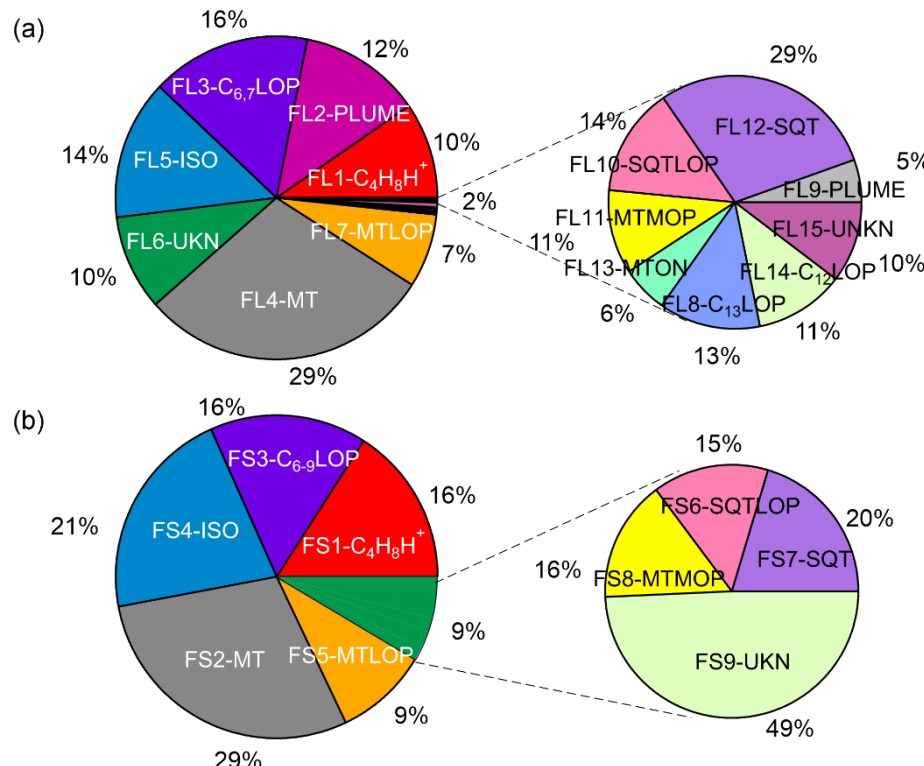

**Figure 12. Average mass contributions of various identified factors in total measured organic vapors (a) in the Landes forest and (b) at the SMEAR II station. The common sources apportioned at both sites are presented in the same color in (a) and (b). FL: factors in the Landes forest; FS: factors at the SMEAR II station; $C_4H_8H^+$, $C_4H_8H^+$ ion-related; PLUME: a plume event; $C_{6,7}$LOP: $C_6$ and $C_7$ lightly oxidized products; $C_{6-9}$LOP: $C_6$-$C_9$ lightly oxygenated compounds; MT: monoterpenes; ISO: isoprene and its oxidation products; UKN: unknown source; MTLOP: monoterpene lightly oxidized products; $C_{12}$LOP: $C_{12}$, $C_{14}$, or $C_{16}$ lightly oxidized products; $C_{13}$LOP: $C_{13}$ lightly oxidized products; SQT: sesquiterpenes; MTON: monoterpene-derived organic nitrates; SQTLOP: sesquiterpene lightly oxidized products; MTMOP: monoterpene more oxidized products.**

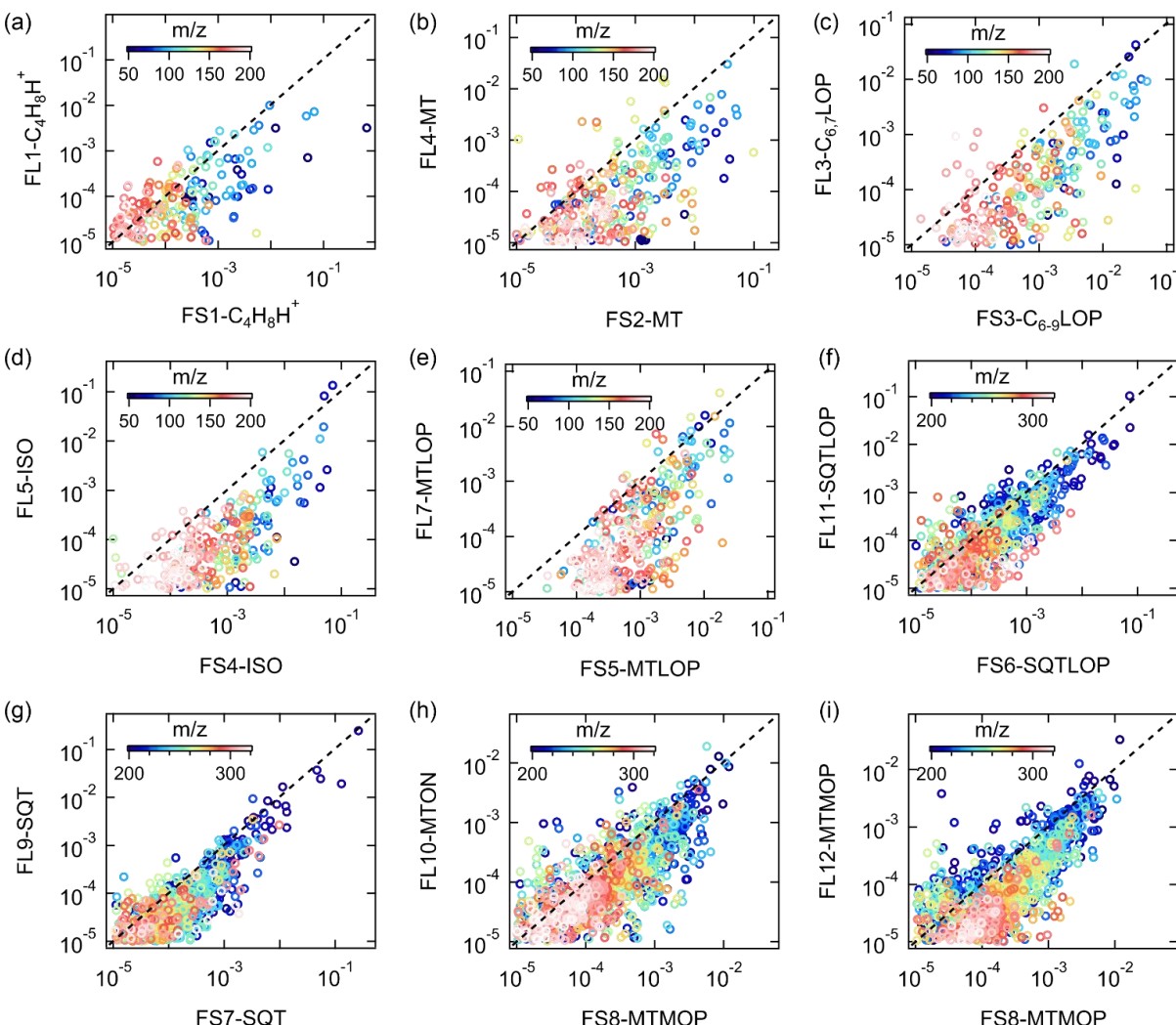

**Figure 13. Comparison between factor profiles of the common sources apportioned in the Landes forest and at the SMEAR II station. The x- and y-axis show the fraction of each bin in the mass spectra of the factors. FL: factors in the Landes forest; FS: factors at the SMEAR II station; $C_4H_8H^+$, $C_4H_8H^+$ ion-related; $C_{6,7}$LOP: MT: monoterpenes; $C_6$ and $C_7$ lightly oxidized products; $C_{6-9}$LOP: $C_6$-$C_9$ lightly oxygenated compounds; ISO: isoprene and its oxidation products; MTLOP: monoterpene lightly oxidized products; $C_{13}$LOP: SQTLOP: sesquiterpene lightly oxidized products; SQT: sesquiterpenes; MTON: monoterpene-derived organic nitrates; MTMOP: monoterpene more oxidized products.**

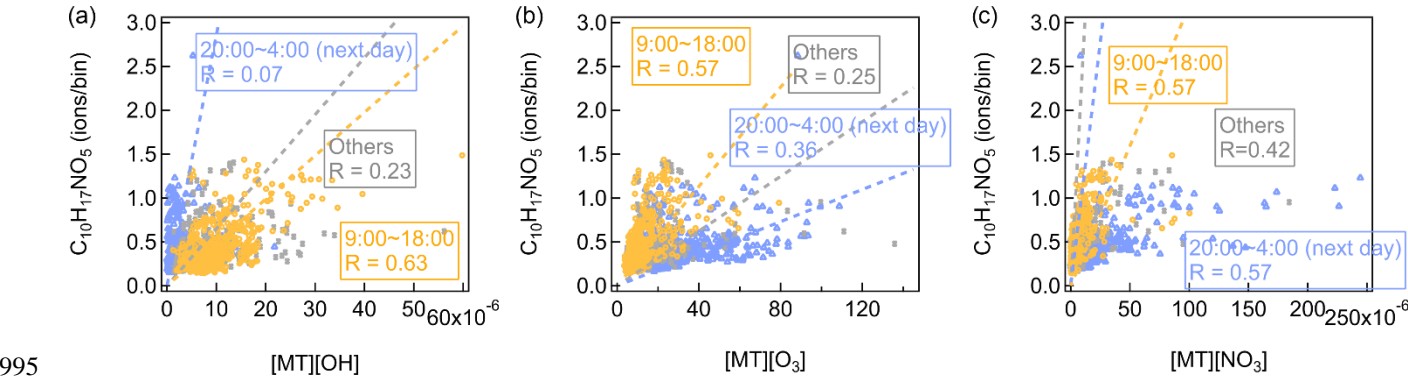

**Figure 14. Scatter plots of $C_{10}H_{17}NO_5$ versus the product of (a) [OH] × [monoterpenes], (b) [O₃] × [monoterpenes], and (c) [NO₃] × [monoterpenes]. Different colours represent different periods of the day.**

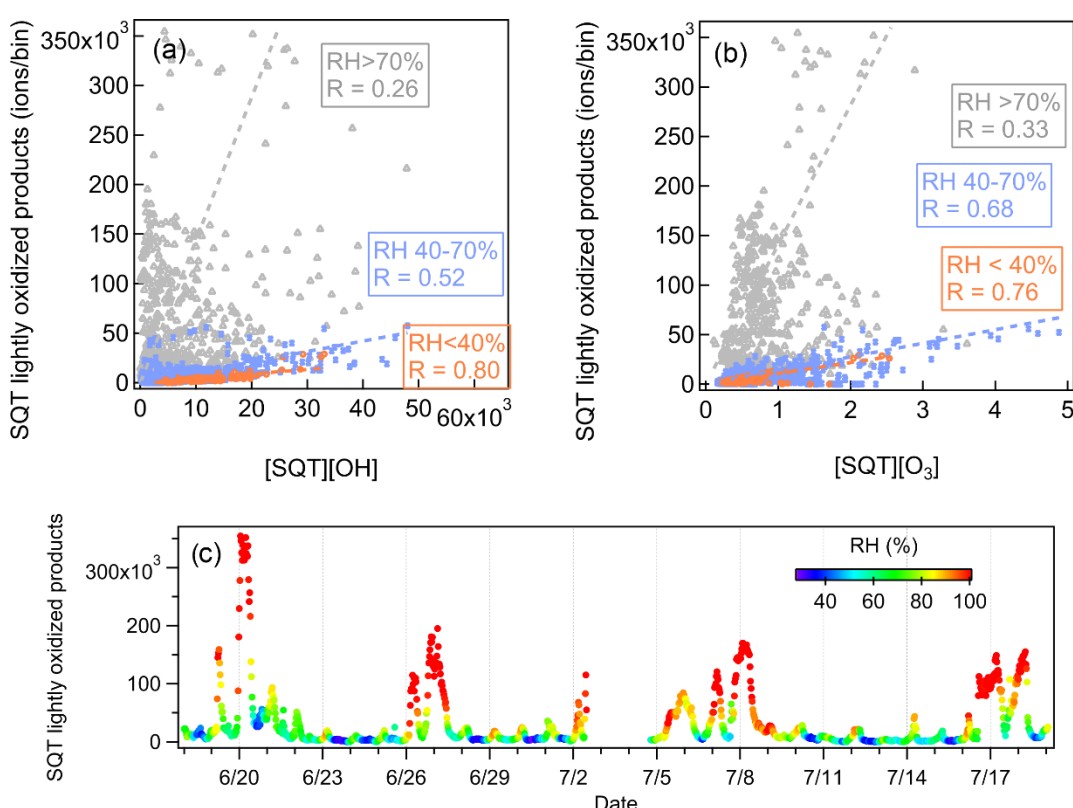

**Figure 15. Scatter plots of sesquiterpene lightly oxidized products versus the product of (a) [OH] × [sesquiterpenes], and (b) [O₃] × [sesquiterpenes]. Different colours indicate different ranges of RH. (c) Time series of sesquiterpene lightly oxidized products colored by RH.**
