# Peer review of "Atmospheric organic vapors in two European pine forests measured by a Vocus PTR-TOF: insights into monoterpene and sesquiterpene oxidation processes"

_Atmospheric Chemistry and Physics, 2020_

## Referee Comment (RC1) · Anonymous Referee #1 · 30 Jul 2020

This article reports the first time the binPMF algorithm has been applied to VOCUS PTR-MS data, as applied to forested environments. There would be a strong interest in this type of broad-base work to try to generalise biogenic emissions, as these can have a profound effect on atmospheric chemistry. While I would say this certainly fits thematically within ACP's scope, right now the paper currently feels unfinished as a research article because while it demonstrates the instrument and algorithm 'working', it currently fails to identify what new understanding this confers to atmospheric science, beyond a running commentary of the authors' interpretations of the factors. I therefore recommend that this paper be published after major revisions. This could take the form of either a research article that is more focused on the atmospheric science arising

from the work, or a technical note that explores the technicalities in more detail (I have queries regarding the methodology, see below). While it could in theory present a 'measurement report' based on this work, I feel that this may not be in the spirit of what the authors intended.

Comments:

The manuscript currently presents the results very systematically, but it is difficult to see what the reader is supposed to get from these. The authors provide a commentary on their interpretation of the factors, but I am not sure I learned anything new or significant about atmospheric chemistry on reading these. If this is to be presented as a research article, the paper needs to be refocused towards the new scientific insight or a testable hypothesis.

A certain amount of work in this paper goes into arriving at factorisations that aren't simply dominated by the big signals. This is probably to be expected because the gas phase VOC ensemble is likely to have many more degrees of freedom than can be accurately represented by the PMF and furthermore, many peaks will have isomers that won't be resolved using PTR. This is done by removing the main monoterpene signal and separating the mass spectrum into different regions. However, this comes across as a little subjective and prevents a direct association between the peaks in the two regions. Did the authors attempt a more conventional approach, such as applying a 'model error' parameter to downweight the larger peaks? More attention should be paid to demonstrating what the effects of not following these procedures in either case, perhaps shown in the supplement.

One might expect that given the number of degrees of freedom available, there will be a level of rotational ambiguity in the solution sets. This certainly would appear to be the case in figure 11, where all of the factors appear to contain traces of siloxane. Was the amount of rotational freedom available explored?

The observation that reaction products did not contribute as much to the mass budget

is perhaps expected because of their chemical lifetime. However, can the authors be sure the these (presumably more polar) molecules were being detected with equal efficiency? Have the authors tried comparing with a mechanistic model like the MCM or GECKO-A?
* * *

---

## Referee Comment (RC2) · Anonymous Referee #2 · 19 Aug 2020

This paper reports on data collected by a VOCUS PTR-ToF-MS at two forest sites. The VOCUS PTR-ToF is a powerful tool to characterize both biogenic and anthropogenic emissions due to it's high sensitivity and broad range of detectable organic compounds. For the first time, the binned positive matrix factorization (binPMF) algorithm has been applied to VOCUS data. Applying PMF to binned data with subsequent high-resolution peak fitting and identification of peaks found to be relevant is a clever way of data reduction in rich datasets as obtained by modern non-selective CIMS techniques.

The paper is technically sound; the authors describe individual PMF factors in great detail, but, unfortunately, the paper does not go beyond a description of observations,

and I agree with Referee #1 that it feels unfinished at this stage. I therefore recommend that this paper be published only after major revisions.

Comments:

I think that the paper does not identify oxidation processes as stated in the abstract, nor does it provide a more comprehensive understanding of gas-phase organic chemistry.

The authors divided the mass spectra into two regions: 51 to 200 Th and 201 to 320 Th. Furthermore, they excluded two masses with high signal intensities (m81 and m137) from the PMF analysis, since these peaks were dominating the mass profiles. As far as I understand, both actions are necessary due to the fact that ambient concentrations of organic species and oxidaion products vary by many orders of magnitude, and the PMF method cannot resolve small signals. Maybe it's worth coming up with either a peak-by-peak normalization method prior to PMF analysis or feed the algorithm with logarithmized signal intensities. Please see this comment being made out of curiosity rather than critical.

Specific comments: Figures 4,7,9 and 12: the y-axis' unit is 'ions/bin' - I think that should be changed into something like 'ions/factor'.

line 62 and 307: replace "complicated" with "complex" line 182: please specifiy what 'high' means. line 293: "much higher intensities" - please rephrase

---

## Author Response (AR1)

**Author response to referee comments**

We thank the referee for the valuable comments which have greatly helped us improve the manuscript. Please find below our responses (in blue) after the referee comments (in black). The changes in the revised manuscript are written in italic.

**Anonymous Referee #1**

This article reports the first time the binPMF algorithm has been applied to VOCUS PTR-MS data, as applied to forested environments. There would be a strong interest in this type of broad-base work to try to generalize biogenic emissions, as these can have a profound effect on atmospheric chemistry. While I would say this certainly fits thematically within ACP's scope, right now the paper currently feels unfinished as a research article because while it demonstrates the instrument and algorithm 'working', it currently fails to identify what new understanding this confers to atmospheric science, beyond a running commentary of the authors' interpretations of the factors. I therefore recommend that this paper be published after major revisions. This could take the form of either a research article that is more focused on the atmospheric science arising from the work, or a technical note that explores the technicalities in more detail (I have queries regarding the methodology, see below). While it could in theory present a 'measurement report' based on this work, I feel that this may not be in the spirit of what the authors intended.

Response: We thank Anonymous Referee #1 for the careful review and inputs which helped improving the overall quality of our work. We agree that as a research article, new understanding towards atmospheric science from this work should be highlighted. Therefore, for the "4 Results and discussion" part, we revised the structure of the section to highlight our major scientific findings and add further analysis to gain insights into the atmospheric processes of monoterpenes and sesquiterpenes.

Compared to preexisting studies, this study performed binPMF analysis on Vocus PTR-TOF data and identified both primary emission sources and secondary oxidation processes of atmospheric organic vapors in two forested environments. For the first time, organic precursors, the lightly oxidized products, and the more oxidized products were separated as individual PMF factors. The relative abundances of these factors can be utilized by modelers to evaluate simulation output, improve model performance, and provide new perspectives to understand gas-phase physicochemical processes. Based on the interpretation of the results relating to oxidation processes, further insights were gained regarding monoterpene and sesquiterpene reactions. For example, a strong relative humidity (RH)-dependence was found for the behavior of sesquiterpene lightly oxidized compounds. High concentrations of these compounds only occur at high RH, yet similar behavior was not observed for monoterpene oxidation products. These findings highlight the need for further studies to delve into gas-phase atmospheric processes of monoterpenes and sesquiterpenes.

More details can be found below as response to the referee's first comment.

Comments:

The manuscript currently presents the results very systematically, but it is difficult to see what the reader is supposed to get from these. The authors provide a commentary on their interpretation of the factors, but I am not sure I learned anything new or significant about atmospheric chemistry on reading these. If this is to be presented as a research article, the paper needs to be refocused towards the new scientific insight or a testable hypothesis.

Response: We agree with the reviewer that as a research article, the new scientific insight or hypothesis should be highlighted to make it clear to the readers. Therefore, the structure of the section "4 Results and discussion" was revised as follows:

"4 Results and discussion

    4.1 Choice of PMF solution and factor interpretation

    4.2 Source identification in the Landes forest

    4.3 Source identification in the southern Finnish boreal forest

    4.4 Comparison among different factors

    4.5 Comparison between the two forests

    4.6 Insights into terpene oxidation processes

        4.6.1 Monoterpene oxidations

        4.6.2 Sesquiterpene oxidations"

In Section 4.4., the identified factors were compared with each other. Based on the similar temporal behavior of Factor L3 ($C_6$ and $C_7$ lightly oxidized products) and Factor L7 ($C_{13}$ lightly oxidized products) and our current knowledge of the corresponding compounds, the $C_{13}$ oxidized compounds are speculated to be produced through the dimer formation mechanisms of $C_6$ and $C_7$ species. The time series of monoterpene lightly oxidized products and sesquiterpene lightly oxidized products do not follow very well with each other, suggesting probably different atmospheric processes. This is further investigated in Section 4.6.

In Section 4.5, spatial comparison between the two forests were discussed regarding the relative abundances of different identified factors. For the common sources identified in both forests, they show similar mass profiles, indicating that the sources and processes are indeed similar despite the quite different regions the forests are in.

In Section 4.6, based on the separation of terpene oxidation processes with varying oxidation degrees, further insights were gained regarding monoterpene and sesquiterpene oxidations (Figure 14, Figure 15, Figure S16, and Figure S17). A strong relative humidity (RH)-dependence was found for the sesquiterpene lightly oxidized compounds, as well as the correlation between them and the products of [OH] × [sesquiterpenes] or [$O_3$] × [sesquiterpenes]. However, these RH dependences were not observed for monoterpene lightly oxidized compounds.

Overall, for the first time, the source identification of atmospheric organic vapors measured by Vocus PTR-TOF separated both primary emission sources and secondary oxidation processes with varying oxidation degrees. The relative abundances of organic precursors, the lightly oxidized products, and the more oxidized products can be utilized by modelers to evaluate simulation output, improve model performance, and provide new perspectives to understand gas-phase physicochemical processes. Based on further investigation of monoterpene and sesquiterpene reactions in the atmosphere, a strong RH-dependence was found 
[revised manuscript text omitted]

A certain amount of work in this paper goes into arriving at factorisations that aren't simply dominated by the big signals. This is probably to be expected because the gas phase VOC ensemble is likely to have many more degrees of freedom than can be accurately represented by the PMF and furthermore, many peaks will have isomers that won't be resolved using PTR. This is done by removing the main monoterpene signal and separating the mass spectrum into different regions. However, this comes across as a little subjective and prevents a direct association between the peaks in the two regions. Did the authors attempt a more conventional approach, such as applying a 'model error' parameter to downweight the larger peaks? More attention should be paid to demonstrating what the effects of not following these procedures in either case, perhaps shown in the supplement.

Response: As the reviewer points out, there are multiple ways in which data can be scaled before factorization, each one giving more or less weight to certain signals in the mass spectra. Earlier studies from our group have explored in detail e.g. scaling according to intensity or mass-to-charge ratio (Fig. 3 and Fig. S7 in Äijälä et al., 2017). The ultimate added value of such labor-intensive approaches is largest when the factorization results are ambiguous or hard to verify. In the current work, we tried a simple approach (removing the main peaks of the largest signals), which will be easy also for others to replicate. This produced factors that made sense both chemically and through their temporal behavior, which lends confidence in the results. The sub-range analysis, which we earlier have shown to be very powerful in separating out less abundant factors (Zhang et al., 2020), also provides a type of "internal verification" when factors with similar temporal and chemical features are resolved

from the two different mass ranges. In the end, there is no single "correct" way to factorize atmospheric data, and the validity of the approach should be referenced to the results, and the conclusion that can be drawn from them.

More specifically concerning this study, the measured signals at $m/z$ 81Th and $m/z$ 137 Th were much higher than the others. In the Vocus PTR-TOF, $m/z$ 81Th mainly comes from the fragmentation of $m/z$ 137 Th (monoterpenes) and therefore follows the characteristics of $m/z$ 137. With the inclusion of these super high peaks (Figure 1a), the mass profiles of three factors were quite similar and dominated by monoterpenes at $m/z$ 137 Th and the major fragment at $m/z$ 81 Th. After exclusion of these high peaks, the mass profiles were more distinct and representative of different factors and at the same time, their temporal behaviors were not interfered (Figure 1b). While the parent ions at $m/z$ 137 Th and $m/z$ 81 were excluded, their corresponding isotopes were retained, effectively downweighting their contributions to the PMF results. The time series of the resolved factors with and without the inclusion of these super high peaks are almost identical. As suggested by the reviewer, the time series and mass profiles of the resolved factors with the inclusion of monoterpene peaks are added in the supplement as Figure S1.

After the exclusion of monoterpene high peaks, if the entire mass spectrum was used for PMF analysis without subranges, factors identified in the high mass range in this study cannot be resolved. As shown in Figure 2, with the entire mass spectrum as PMF input, most identified factors in the low mass range were resolved although there were some mixing of different factors. For example, the factors of $C_6$ and $C_7$ lightly oxidized products, a plume event, monoterpenes, unknown source, monoterpene lightly oxidized products, and isoprene and its oxidation products, were clearly seen. However, the PMF analysis cannot separate the factors of sesquiterpenes, sesquiterpene lightly oxidized products, monoterpene more oxidized products, monoterpene-derived organic nitrates, and $C_{13}$ lightly oxidized products. Increasing the number of factors for PMF run did not help.

In this study, with the factorization on subranges of the mass spectra, different factors representing primary emission sources and secondary oxidation processes were identified in both mass ranges. The association between these two ranges were further explored by comparison of their time series, diurnal variations, and correlation analysis (Figure 11 in the manuscript). For example, the factors of a plume event were resolved in both mass ranges and their time series correlated closely with each other. The monoterpene factor in the low mass range showed a good correlation with the sesquiterpene factor in the high mass range. Interestingly, the factor of $C_6$ and $C_7$ lightly oxidized products in the low mass range correlated very well with the factor of $C_{13}$ lightly oxidized products in the high mass range, which lead to the speculation that the $C_{13}$ oxygenated compounds are produced through the dimer formation mechanisms of $C_6$ and $C_7$ species. In addition, the factor of monoterpene lightly oxidized products showed a poor correlation with the factor of sesquiterpene lightly oxidized products. Without the PMF analysis on subranges of mass spectra, these factors and different processes cannot be separated. Zhang et al. (2020) performed factor analysis on subranges of mass spectra measured by $NO_3^-$ CIMS, and found that the formation of daytime dimer and the monoterpene dimers from the combined products of $NO_3$ and $O_3$ oxidations cannot be resolved without the subrange approach.

[Figure]

Figure 1. The mass profiles and time series of the seven-factor solution for the low mass range in the Landes forest (a) with and (b) without the inclusion of the signals at *m/z* 81 Th and *m/z* 137 Th.

[Figure]

Figure 2. The mass profiles and time series of the eight-factor solution in the Landes forest with the entire mass spectrum as input of PMF analysis. We varied the FPEAK value between -1 and +1 with the step of 0.2. Taking the high mass range of 201-320 Th at the SMEAR IIstation as the example,

One might expect that given the number of degrees of freedom available, there will be a level of rotational ambiguity in the solution sets. This certainly would appear to be the case in figure 11, where all of the factors appear to contain traces of siloxane. Was the amount of rotational freedom available explored?

Response: The rotational freedom of the PMF solutions in this study was explored through use of the FPEAK parameters. For the optimal solutions, we varied the FPEAK value between -1 and +1 with the step of 0.2. For the low mass range of 51-200 Th of the Landes and SMEAR II dataset, the variations in FPEAK value did not influence the mass profile and time series much. For the high mass range of 201-320 Th, we saw the changes especially in the factor profiles by varying FPEAK values. For the Landes measurements, Figure 3 shows the factor profiles of the eight-factor solution with FPEAK = 0, +0.6, and -0.6. The time series of different factors for these FPEAK values are similar. After a detailed evaluation, we found no evidence that solutions with FPEAK value away from zero are preferable. However, for the high mass range of the SMEAR II measurements, as expected by the reviewer, the solutions with positive values of FPEAK work better than that with FPEAK = 0 in terms of the factor profiles. As shown in Fig. 4, by varying FPEAK with positive values, the factor profile of monoterpene more oxidized products (including organic nitrates) contained less traces of siloxanes and showed elevated fractions of the fingerprint peaks. After evaluation, we decided to choose the solution with FPEAK = +0.6 for the high mass range of the SMEAR II dataset.

The corresponding information of rotational ambiguity has been added in the revised manuscript (Lines 230-239): *"The rotational freedom of the PMF solutions was explored through the use of the FPEAK parameters. For each of the optimal solutions, we varied the FPEAK values between -1 and +1 with the step of 0.2. For the low mass ranges of the Landes and SMEAR II dataset, the varying FPEAK values did not change the factor profiles and time series much. For the high mass range of the Landes measurements, we saw variations especially in the factor profiles by varying FPEAK values. But after a detailed evaluation, we found no evidence that solutions with*

*FPEAK values away from zero were preferable. However, for the high mass range of the SMEAR II measurements, the solutions with positive values of FPEAK worked better than that with FPEAK = 0 in terms of factor profiles. The factor time series were similar when FPEAK values varied. But for the factor profiles with positive FPEAK values, the factor of monoterpene more oxidized products including organic nitrates contained less traces of siloxanes and showed elevated fractions of the corresponding fingerprint peaks (Fig. S12). After evaluation, we chose the solution with FPEAK = +0.6 for the high mass range of the SMEAR II dataset.*"

Figure 9, Figure 10, Figure 12, and Figure 13 have been updated accordingly.

[Figure]

Figure 3. The factor profiles of the eight-factor solution for the high mass range of the Landes measurements with FPEAK = 0, +0.6, and -0.6.

[Figure]

Figure 4. The factor profiles of the four-factor solution for the high mass range of the SMEAR II measurements with FPEAK = 0, +0.6, and -0.6.

The observation that reaction products did not contribute as much to the mass budget is perhaps expected because of their chemical lifetime. However, can the authors be sure the these (presumably more polar) molecules were

being detected with equal efficiency? Have the authors tried comparing with a mechanistic model like the MCM or GECKO-A?

Response: The sensitivities of different VOCs in the PTR instrument are not equal and are linearly related to their proton-transfer reaction rate constants when ion transmission efficiency and fragmentation ions are considered (Sekimoto et al., 2017; Krechmer et al., 2018). According to Sekimoto et al. (2017), the reaction rate constants of different molecules significantly depend on their molecular mass, elemental composition, and functionality. In this study, we acknowledge that it is not a perfect method to quantify the mass fraction of different factors based on their average signal intensities as shown in the pie charts of Figure 12. The related uncertainties are discussed in the manuscript (Lines 419-423): "We acknowledge that it is not a perfect method to quantify the contributions of various sources and formation processes. The sensitivities of different VOCs measured by the PTR instruments may vary by a factor of 2-3 (Sekimoto et al., 2017; Yuan et al., 2017). The uncertainties can come from the challenge to convert the signal intensity to atmospheric concentrations because of problematic calibrations, especially given that many unknown molecules exist in the mass spectra."

In this study, a large mass fraction of the gas-phase organic species were measured and classified including the precursors, the lightly oxidized products, and the more oxidized products, which was not achieved by previous studies. Although it is out of the scope of the current study to perform model simulations, our results provide good data base for potential model study in the future to compare model simulations with our ambient observations, improve model performance, and help scientists better understand the complex atmospheric chemistry. Still, the lack of speciation of e.g. the monoterpenes with the PTR approach remains a challenge for mechanistic modeling, as the oxidation product distributions will vary tremendously depending on the exact VOC distributions in the forests.

**Anonymous Referee #2**

This paper reports on data collected by a VOCUS PTR-ToF-MS at two forest sites. The VOCUS PTR-ToF is a powerful tool to characterize both biogenic and anthropogenic emissions due to it's high sensitivity and broad range of detectable organic compounds. For the first time, the binned positive matrix factorization (binPMF) algorithm has been applied to VOCUS data. Applying PMF to binned data with subsequent high-resolution peak fitting and identification of peaks found to be relevant is a clever way of data reduction in rich datasets as obtained by modern non-selective CIMS techniques.

The paper is technically sound; the authors describe individual PMF factors in great detail, but, unfortunately, the paper does not go beyond a description of observations, and I agree with Referee #1 that it feels unfinished at this stage. I therefore recommend that this paper be published only after major revisions.

Response: We thank Anonymous Referee #2 for the careful review and inputs which helped improving the overall quality of our work. We agree that as a research article, the paper should go beyond a description of PMF source apportionment and highlight new understanding towards atmospheric science from this work. Therefore, in the revised manuscript, our major findings are highlighted and more insights are gained into monoterpene and sesquiterpene oxidations. Please see more details in our responses to Referee #1.

Comments:

I think that the paper does not identify oxidation processes as stated in the abstract, nor does it provide a more comprehensive understanding of gas-phase organic chemistry.

Response: In this study, a large mass fraction of the gas-phase organic species were measured and classified. In addition to the precursors, their lightly oxidized products and more oxidized products were separated as individual factors. Based on the interpretation of these factors related to oxidation processes, further insights were gained regarding monoterpene and sesquiterpene reactions. In addition, the relative abundances of organic precursors, the lightly oxidized products, and the more oxidized products can be utilized by modelers to evaluate simulation output, improve model performance, and provide new perspectives to understand gas-phase physicochemical processes.

We revised the abstract as follows:

*"Atmospheric organic vapors play essential roles in the formation of secondary organic aerosol. Source identification of these vapors is thus fundamental to understand their emission sources and chemical evolution in the atmosphere and their further impact on air quality and climate change. In this study, a Vocus proton-transfer-reaction time-of-flight mass spectrometer (PTR-TOF) was deployed in two forested environments, the Landes forest in southern France and the boreal forest in southern Finland, to measure atmospheric organic vapors, including both volatile organic compounds (VOCs) and their oxidation products. For the first time, we performed binned positive matrix factorization (binPMF) analysis on the complex mass spectra acquired with the Vocus PTR-TOF and identified various emission sources as well as oxidation processes in the atmosphere. Based on separate analysis of low- and high-mass ranges, fifteen PMF factors in the Landes forest and nine PMF factors in the Finnish boreal forest were resolved, showing a high similarity between the two sites. Particularly, terpenes and various terpene reaction products were separated into individual PMF factors with varying oxidation degrees, such as lightly oxidized compounds from both monoterpene and sesquiterpene oxidations, monoterpene-derived organic nitrates, and monoterpene more oxidized compounds. Factors representing monoterpenes dominated the biogenic VOCs in both forests, with less contributions from the isoprene factors and sesquiterpene factors. Factors of the lightly oxidized products, more oxidized products, and organic nitrates of monoterpenes/sesquiterpenes accounted for 8-12% of the measured gas-phase organic vapors in the two forests. Based on the interpretation of the results relating to oxidation processes, further insights were gained regarding monoterpene and sesquiterpene reactions. For example, a strong relative humidity (RH)-dependence was found for the behavior of sesquiterpene*

*lightly oxidized compounds. High concentrations of these compounds only occur at high RH, yet similar behavior was not observed for monoterpene oxidation products. These findings highlight the need for further studies to delve into gas-phase atmospheric processes of monoterpenes and sesquiterpenes."*

The authors divided the mass spectra into two regions: 51 to 200 Th and 201 to 320 Th. Furthermore, they excluded two masses with high signal intensities (m81 and m137) from the PMF analysis, since these peaks were dominating the mass profiles. As far as I understand, both actions are necessary due to the fact that ambient concentrations of organic species and oxidaion products vary by many orders of magnitude, and the PMF method cannot resolve small signals. Maybe it's worth coming up with either a peak-by-peak normalization method prior to PMF analysis or feed the algorithm with logarithmized signal intensities. Please see this comment being made out of curiosity rather than critical.

Response: As the reviewer points out, there are multiple ways in which data can be scaled before factorization, each one giving more or less weight to certain signals in the mass spectra. Earlier studies from our group have explored in detail e.g. scaling according to intensity or mass-to-charge ratio (Fig. 3 and Fig. S7 in Äijälä et al., 2017). The ultimate added value of such labor-intensive approaches is largest when the factorization results are ambiguous or hard to verify. In the current work, we tried a simple approach (removing the main peaks of the largest signals), which will be easy also for others to replicate. This produced factors that made sense both chemically and through their temporal behavior, which lends confidence in the results. The sub-range analysis, which we earlier have shown to be very powerful in separating out less abundant factors (Zhang et al., 2020), also provides a type of "internal verification" when factors with similar temporal and chemical features are resolved from the two different mass ranges. In the end, there is no single "correct" way to factorize atmospheric data, and the validity of the approach should be referenced to the results, and the conclusion that can be drawn from them.

Specific comments: Figures 4,7,9 and 12: the y-axis' unit is 'ions/bin' - I think that should be changed into something like 'ions/factor'.

Response: These figures show the time series of different factors. The unit corresponds to the binned signal intensities measured by the mass spectrometer and should be "ions/bin".

line 62 and 307: replace "complicated" with "complex"

Response: Replaced.

line 182: please specifiy what 'high' means.

Response: As shown in Figure S2, for some bins, the scaled residual can as high as ±200. In the revised manuscript (Line 196), it is specified as "*For some bins the residuals are still high (the scaled residuals as high as ±200).*"

line 293: "much higher intensities" - please rephrase

Response: Done.

Correspondence: Haiyan Li (haiyan.li@helsinki.fi)

**Abstract.**

Atmospheric organic vapors play essential roles in the formation of secondary organic aerosol. Source identification of these vapors is thus fundamental to understand their emission sources and chemical evolution in the atmosphere and their further impact on air quality and climate change. In this study, a Vocus proton-transfer-reaction time-of-flight mass spectrometer
20 (PTR-TOF) was deployed in two forested environments, the Landes forest in southern France and the boreal forest in southern Finland, to measure atmospheric organic vapors, including both volatile organic compounds (VOCs) and their oxidation products. For the first time, we performed binned positive matrix factorization (binPMF) analysis on the complex mass spectra acquired with the Vocus PTR-TOF and identified various emission sources as well as oxidation processes in the atmosphere. Based on separate analysis of low- and high-mass ranges, fifteen PMF factors in the Landes forest and nine PMF factors in the
25 Finnish boreal forest were resolved, showing a high similarity between the two sites. Particularly, terpenes and various terpene reaction products were separated into individual PMF factors with varying oxidation degrees, such as lightly oxidized compounds from both monoterpene and sesquiterpene oxidations, monoterpene-derived organic nitrates, and monoterpene more oxidized compounds. Factors representing monoterpenes dominated the biogenic VOCs in both forests, with less contributions from the isoprene factors and sesquiterpene factors. Factors of the lightly oxidized products, more oxidized
30 products, and organic nitrates of monoterpenes/sesquiterpenes accounted for 8-12% of the measured gas-phase organic vapors in the two forests. Based on the interpretation of the results relating to oxidation processes, further insights were gained regarding monoterpene and sesquiterpene reactions. For example, a strong relative humidity (RH)-dependence was found for the behavior of sesquiterpene lightly oxidized compounds. High concentrations of these compounds only occur at high RH,

yet similar behavior was not observed for monoterpene oxidation products. These findings highlight the need for further studies
35 to delve into gas-phase atmospheric processes of monoterpenes and sesquiterpenes. ~~
[revised manuscript text omitted]

[Figure]

Figure S21. The distribution of scaled residuals as a function of *m/z* of the seven-factor solution for the low mass range in the Landes forest.

[Figure]

Figure S32. The six-factor solution for the low mass range in the Landes forest, showing (a) factor mass profiles, (b) factor time series, and (c) diurnal cycles of different factors.

[Figure]

Figure S43. The eight-factor solution for the low mass range in the Landes forest, showing (a) factor mass profiles, (b) factor time series, and (c) diurnal cycles of different factors.

[Figure]

Figure S54. The distribution of scaled residuals as a function of *m/z* of the eight-factor solution for the high mass range in the Landes forest.

[Figure]

Figure S65. The seven-factor solution for the high mass range in the Landes forest, showing (a) factor mass profiles, (b) factor time series, and (c) diurnal trends of different factors.

[Figure]

Figure S76. The nine-factor solution for the high mass range in the Landes forest, showing (a) factor mass profiles, (b) factor time series, and (c) diurnal trends of different factors.

[Figure]

Figure S8̶7̶. The four-factor solution for the low mass range at SMEAR II station, showing (a) factor mass profiles, (b) factor time series, and (c) diurnal trends of different factors.

[Figure]

Figure S98. The six-factor solution for the low mass range at SMEAR II station, showing (a) factor mass profiles, (b) factor time series, and (c) diurnal trends of different factors.

[Figure]

Figure S10. The three-factor solution for the high mass range at SMEAR II station, showing (a) factor mass profiles, (b) factor time series, and (c) diurnal cycles of different factors.

[Figure]

Figure S110. The five-factor solution for the high mass range at SMEAR II station, showing (a) factor mass profiles, (b) factor time series, and (c) diurnal cycles of different factors.

[Figure]

Figure S12. The factor profiles of the four-factor solution for the high mass range of the SMEAR Ⅱ measurements with FPEAK = 0, +0.6, and -0.6.

[Figure]

Figure S13. Correlations between PMF factors and marker molecules in the Landes forest, with the color representing the correlation coefficients ($r^2$).

[Figure]

Figure S11. The correlations among various factors identified in the Landes forest, with the color representing the correlation coefficients ($r^2$).

[Figure]

Figure S12. Bivariate polar plot of $C_4H_9^+$ signal measured at SMEAR II station as a function of wind speed and wind direction using the OpenAir software (Carslaw and Ropkins, 2012).

[Figure]

Figure S14. Correlations between PMF factors and marker molecules at the SMEAR II station, with the color indicating the correlation coefficients ($r^2$).

[Figure]

Figure S13. The correlations among various factors identified at SMEAR II station, with the color representing the correlation coefficients ($r^2$).

[Figure]

Figure S15̵2. Bivariate polar plot of $C_4H_9^+$ signal measured at SMEAR II station as a function of wind speed and wind direction using the OpenAir software (Carslaw and Ropkins, 2012).

[Figure]

Figure S16. Scatter plots of $C_{10}H_{15}NO_6$ versus the product of (a) [OH] × [monoterpenes], (b) [O_3] × [monoterpenes], and (c) [NO_3] × [monoterpenes]. Different colours represent different periods of the day.

[Figure]

Figure S17. Scatter plots of monoterpene lightly oxidized products versus the product of (a) [OH] × [monoterpenes], and (b) [O$_3$] × [monoterpenes]. Different colours indicate different ranges of RH. (c) Time series of monoterpene lightly oxidized products colored by RH.

**References**

Carslaw, D. C. and Ropkins, K.: openair – An R package for air quality data analysis, Environ. Modell. Softw., 27–28, 52–61, 2012.

---

## Author Response (AR2)

**Author response to referee comments**

Comments to the Author:

I thank the authors for considering my comments and I feel now that the paper is more in-scope regarding the discussion of potential new insights into atmospheric chemistry. I'm more happy for this to be published in ACP, however I do have some comments that I would like to see addressed before publication.

Response: We thank the reviewer for the positive and constructive reviews of our revision. Below, we go through them point by point, with the original remarks in black and our replies in blue. The changes in the revised manuscript are in italic.

General comments:

While the authors discuss in more detail the reasoning behind their method to exclude the larger peaks, I do not think they pay it enough attention to it in the revised manuscript. It is not sufficient to simply demonstrate that a factorisation looks unsatisfactory; the authors should discuss the reasons why it behaves the way it does, and better justify their chosen remedy in this context.

Response: We agree with the reviewer that more discussions should be added in the revised manuscript rather than only in the response. In the revised manuscript, we explained why the mass profiles of several resolved factors look quite similar with inclusion of the large monoterpene-related peaks, which leads to unsatisfactory results. Also we justified that our simple approach (removing the main peaks of the largest signals) worked very well in this study and produced factors that made sense both chemically and through their time variations.

The discussions have been added in the revised manuscript (Lines 155-161):

*"Because PMF assumes that the data matrix can be explained by a linear combination of different factors, even a very tiny fraction of these high peaks was split into a factor, they may dominate the mass profile of the factor. As shown in Fig. S1, with the inclusion of $C_6H_8H^+$ and $C_{10}H_{16}H^+$, the mass profiles of several factors were quite similar and dominated by these peaks. Therefore, the major mass bins of these ions were excluded for further PMF analysis, but their corresponding isotopes were retained, effectively downweighting their contributions to the PMF result. This simple approach by removing the main peaks of the largest signals produced factors that made sense both chemically and through their temporal behaviors, lending confidence in the results."*

I thank the authors for taking on board my comment exploring rotational ambiguity and the use of FPEAK to explore this is certainly informative in some ways. However, I could question the following concerning the approach:

1. If little difference is found across the range of FPEAK values, then this can simply mean that FPEAK has not been varied across a sufficient range, noting that the effect of varying this parameter will vary dataset to dataset.

Response: We agree with the reviewer that the effect of varying FPEAK values varies dataset to dataset and different ranges of FPEAK have been explored in previous studies (Ulbrich et al., 2009; Zhang et al., 2011). In this study, we follow solutions generally reported in the literature with an FPEAK value between -1 and +1 (Reff et al., 2007). For the low mass ranges of the Landes and SMEAR II dataset, the changes in the factor profiles and time series were very small by varying FPEAK values, indicating that varying FPEAK values from -1 to +1 did not affect the overall results of PMF analysis.

In the revised manuscript, we added in Lines 239-240 that "*For the low mass ranges of the Landes and SMEAR II dataset, the varying FPEAK values did not change the factor profiles and time series much, indicating that varying FPEAK values from -1 to +1 did not affect the overall results of PMF analysis.*"

Ulbrich, I. M., Canagaratna, M. R., Zhang, Q., Worsnop, D. R., and Jimenez, J. L.: Interpretation of organic components from Positive Matrix Factorization of aerosol mass spectrometric data, Atmos. Chem. Phys., 9, 2891-2918, 10.5194/acp-9-2891-2009, 2009.

Zhang, Q., Jimenez, J. L., Canagaratna, M. R., Ulbrich, I. M., Ng, N. L., Worsnop, D. R., Sun, Y. J. A., and Chemistry, B.: Understanding atmospheric organic aerosols via factor analysis of aerosol mass spectrometry: a review, 401, 3045-3067, 10.1007/s00216-011-5355-y, 2011.

Reff, A., Eberly, S. I., and Bhave, P. V.: Receptor Modeling of Ambient Particulate Matter Data Using Positive Matrix Factorization: Review of Existing Methods, J Air Waste Manage, 57, 146-154, 10.1080/10473289.2007.10465319, 2007.

2. The authors seem to basing their choice of FPEAK on the level of separation of mass spectral profiles. I would note that such an approach will tend to favour the positive values of FPEAK, however this does not necessarily make the factors more 'real' because one would expect a certain amount of overlap between the factors in mass spectral space. The authors should explain this better (the assumption that siloxane should only feature in one factor could be considered reasonable, but the reasoning is left largely implicit).

Response: Changing FPEAK changes both the mass profiles and time series of PMF factors simultaneously. In this study, we compared both factor profiles and factor time series to choose a more optimal FPEAK value. For the high mass range of the SMEAR II dataset, the factor time series were similar with FPEAK values from -1 to +1. However, for PMF solutions with positive FPEAK values, the factor profile of monoterpene more oxidized products including organic nitrates contained less traces of siloxanes and was more dominated by the corresponding fingerprint peaks. As discussed in section 4.3, these siloxanes can come from cosmetics and personal care products, and silicone oils used in instrument pumps. The temporal variations of these siloxanes showed high background signals and presented regular diurnal cycles following the variations in ambient temperature, which were significantly different from those of monoterpene more oxidized products. We believe that these siloxanes come from or represent a much different source from monoterpene more oxidized products. Therefore, we chose the PMF solution of FPEAK = +0.6 where siloxanes featured more in one single factor.

As the reviewer suggested, we add more explanations in the revised manuscript (Lines 246-249): *"As discussed in Sect. 4.3, these siloxanes can come from cosmetics and personal care products, and silicone oils used in instrument pumps. The temporal variations of these siloxanes differed significantly from those of monoterpene more oxidized products. After evaluation, we chose the solution with FPEAK = +0.6 for the high mass range of the SMEAR II dataset, where siloxanes feature more in one factor."*

3. There is an explicit recommendation in Paatero (2002) that nonzero FPEAK values should not normally be used for environmental data, so using this as the 'final' factorisation needs to be justified in this context. To be completely clear, my recommendation in the first review was to merely explore (and hopefully quantify) the rotational ambiguity, not necessarily to choose a more optimal FPEAK value.

Response: It has been demonstrated by Hopke (2000) that for real datasets, positive, non-zero FPEAK values generally yield more realistic PMF results. In this study, we explored the rotational ambiguity of PMF results using FPEAK values and found that factorization with positive FPEAK values produce more reasonable PMF solutions. The justification for choosing the PMF solution of FPEAK = +0.6 has been discussed above and added in the revised manuscript.

Hopke, P. K.: A guide to positive matrix factorization, Workshop on UNMIX and PMF as Applied to PM2.5, Research Triangle Park, NC, USA, 2000.

Specific comments:

Line 194: How are the authors defining 'significant'?

Response: After seven factors, the step change in $Q/Q_{exp}$ is lower than 7%. We make the use of "significant" quantitatively in the revised manuscript (Line 198).

Line 195: A more explicit description of the criteria used to choose the number of factors is needed.

Response: To choose the optimum number of factors, we examined multiple criteria including the variations of $Q/Q_{exp}$ vs. varying factor number, the distribution of the scaled residuals for each $m/z$, sum of the squares of scaled residuals, factor mass profile, factor time series and diurnal cycles, and also signs of split factors, following the procedures proposed by Ulbrich et al. (2009) and Zhang et al. (2011).

We include the description of these criteria in the revised manuscript (Lines 199-201): *"The optimal solution of seven factors was chosen by evaluating the variations of $Q/Q_{exp}$ vs. varying factor number, the distribution of the scaled residuals for each m/z, sum of the squares of scaled residuals, factor mass profile, factor time series and diurnal cycles, and also signs of split factors (Ulbrich et al., 2009; Zhang et al., 2011)."*

Line 493: Regarding the two routes to nitrate formation, the caveat should be included that just because they are contained in the same factor, it does not mean that the same molecules are being produced. With the current wording, a reader could be mislead into thinking this to be the case.

Response: As suggested by the reviewer, the caveat has been included in the revised manuscript (Lines 503-505): *"It should be noted that $C_{10}H_{17}NO_5$ and $C_{10}H_{15}NO_6$ are used as examples because both of them are fingerprint peaks of the factor, but in real environments it may not be the case that these molecules are always produced from the above formation routes."*

Line 517: The end of this section is rather unsatisfactory, leaving the reader wondering whether the observation is linked to chemistry, time of day, insolation, temperature, etc. The authors make speculations regarding day vs night and sunny days, so can these be tested? As a bare minimum, the authors should (1) rule out temperature as a controlling factor because this is strongly anticorrelated with humidity and is known to influence reaction rates, branching ratios, etc., and (2) attempt to see if nocturnal nitrate chemistry could be responsible.

Response: The reason behind the high RH-dependence of sesquiterpene lightly oxidized products cannot be figured out based on the measurements in this study. But as the reviewer suggested, the controlling role of temperature should be ruled out because temperature is strongly anti-correlated with RH and is known to influence terpene emissions and terpene reaction rates. The strong RH-dependence was observed for sesquiterpene lightly oxidized products where sesquiterpene-derived organic nitrates are not included (not observed in this study). Therefore, we do not think nocturnal nitrate chemistry could be the reason.

In the revised manuscript, we add in Lines 528-530 that "*The controlling role of temperature can be ruled out because temperature is strongly anti-correlated with RH and is known to influence terpene emissions and terpene reaction rates.*"

Technical corrections:

Line 33: The comment "These findings highlight the need for further studies to delve into gas-phase atmospheric processes of monoterpenes and sesquiterpenes" verges on pointing out the obvious, as terpene chemistry has been a major topic of atmospheric chemistry for decades.

Response: This comment has been removed.

Line 61: Replace 'revealed' with 'showed'

Response: Done.

Line 64: Remove 'state-of-the-art'. Given that two authors have a financial interest in the instrument, it is important that the language describing it remains objective.

Response: Removed.

Line 468 (and elsewhere): Replace "oxidations" with "oxidation"

Response: Done.

[revised manuscript text omitted]